# Enhanced excitability of cortical neurons in low-divalent solutions is primarily mediated by altered voltage-dependence of voltage-gated sodium channels

Briana J Martiszus[1,2], Timur Tsintsadze[1,2], Wenhan Chang[3], Stephen M Smith[1,2]*

[1]Section of Pulmonary & Critical Care Medicine, VA Portland Health Care System, Portland, United States; [2]Department of Medicine, Division of Pulmonary & Critical Care Medicine, Oregon Health & Science University, Portland, United States; [3]Endocrine Research Unit, Veterans Affairs Medical Center and University of California, San Francisco, San Francisco, United States

**Abstract** Increasing extracellular [Ca2+] ([Ca2+]o) strongly decreases intrinsic excitability in neurons but the mechanism is unclear. By one hypothesis, [Ca2+]o screens surface charge, reducing voltage-gated sodium channel (VGSC) activation and by another [Ca2+]o activates Calcium-sensing receptor (CaSR) closing the sodium-leak channel (NALCN). Here we report that neocortical neurons from CaSR-deficient (Casr-/-) mice had more negative resting potentials and did not fire spontaneously in reduced divalent-containing solution (T0.2) in contrast with wild-type (WT). However, after setting membrane potential to $-70$ mV, T0.2 application similarly depolarized and increased action potential firing in Casr-/- and WT neurons. Enhanced activation of VGSCs was the dominant contributor to the depolarization and increase in excitability by T0.2 and occurred due to hyperpolarizing shifts in VGSC window currents. CaSR deletion depolarized VGSC window currents but did not affect NALCN activation. Regulation of VGSC gating by external divalents is the key mechanism mediating divalent-dependent changes in neocortical neuron excitability.

*For correspondence:
smisteph@ohsu.edu

Competing interests: The authors declare that no competing interests exist.

## Introduction

Excitable tissues are strongly regulated by extracellular [Ca$^{2+}$] ([Ca$^{2+}$]$_o$) (*Neher and Sakaba, 2008*; *Ma et al., 2014*; *Jackman and Regehr, 2017*). Movement of extracellular Ca$^{2+}$, through voltage-activated Ca$^{2+}$ channels (VACC), to the intracellular space is central to many of these processes (*Ma et al., 2012a*; *Nanou and Catterall, 2018*). However, a distinct, extracellular mechanism that is independent of synaptic transmission also contributes to [Ca$^{2+}$]$_o$-dependent regulation of nerve and muscle function (*Adrian and Gelfan, 1933*; *Weidmann, 1955*; *Frankenhaeuser, 1957*; *Frankenhaeuser and Hodgkin, 1957*). Decreases in [Ca$^{2+}$]$_o$ and [Mg$^{2+}$]$_o$ substantially facilitate spontaneous and evoked action potential generation which represents increased intrinsic excitability (*Weidmann, 1955*; *Frankenhaeuser, 1957*; *Frankenhaeuser and Hodgkin, 1957*). In the brain, physiological neuronal activity decreases [Ca$^{2+}$]$_o$ (*Nicholson et al., 1978*) leading to further increases in action potential firing in neighboring neurons (*Anderson et al., 2013*). The firing patterns and computational properties of local circuits are impacted substantially by this positive feedback leading to changes in brain behaviors (*Titley et al., 2019*). Furthermore, under pathological conditions, larger decreases in [Ca$^{2+}$]$_o$ occur, resulting in even greater changes in circuit activity, and implicating [Ca$^{2+}$]$_o$-dependent excitability in the pathogenesis of brain injury (*Ayata and Lauritzen, 2015*).

Classical studies proposed that the mechanism underlying [Ca$^{2+}$]$_o$-dependent excitability centers on voltage-gated sodium channel (VGSC) sensitivity to extracellular Ca$^{2+}$. Reduced [Ca$^{2+}$]$_o$ was

proposed to shift the effective voltage-dependent gating of the sodium conductance in the hyper-polarizing direction by reducing the screening of local negative charges on the extracellular face of the membrane or channel by external $Ca^{2+}$ (*Frankenhaeuser and Hodgkin, 1957*; *Hille, 1968*). This surface potential screening model accounted for $[Ca^{2+}]_o$-dependent excitability in nerves and muscle without a need for additional molecular players and was widely accepted (*Hille, 2001*), although direct binding of $Ca^{2+}$ to the VGSC was also proposed as contributing (*Armstrong and Cota, 1991*). However, this theory was challenged by new data demonstrating that activation of the sodium leak channel (NALCN), a non-selective cation channel, by the intracellular proteins, UNC79 and UNC80 (*Lu et al., 2009*; *Lu et al., 2010*) was necessary for $[Ca^{2+}]_o$-dependent excitability to occur in hippocampal neurons. Following the deletion of NALCN or UNC79, $[Ca^{2+}]_o$-dependent excitability was completely lost suggesting the increased excitability resulted from the activation of the non-rectifying NALCN which depolarized neurons and increased the likelihood of action potential generation *independent* of changes in VGSC function (*Lu et al., 2010*). The calcium-sensing receptor (CaSR), a G-protein-coupled receptor (GPCR), was hypothesized to detect and transduce the $[Ca^{2+}]_o$ changes and signal to the downstream multistep pathway (*Lu et al., 2010*). CaSR is well-positioned as a candidate $[Ca^{2+}]_o$ detector because at nerve terminals it detects $[Ca^{2+}]_o$ and regulates a non-selective cation channel (*Smith et al., 2004*; *Chen et al., 2010*) and because it transduces changes in $[Ca^{2+}]_o$ into NALCN activity following heterologous co-expression of CaSR, NALCN, UNC79, and UNC80 (*Lu et al., 2010*). Interest in the UNC79-UNC80-NALCN pathway has also risen, due to its essential role in the maintenance of respiration (*Lu et al., 2007*), the regulation of circadian rhythms (*Lear et al., 2013*; *Flourakis et al., 2015*), and because mutations of UNC80 and NALCN cause neurodevelopmental disorders, characterized by development delay and hypotonia (*Al-Sayed et al., 2013*; *Perez et al., 2016*).

Here, we address the question of whether the G-protein mediated NALCN pathway or VGSCs transduce the $[Ca^{2+}]_o$-dependent effects on excitability. We test if CaSR is a modulator of neuronal excitability via its action on a nonselective cation channel, determine the impact of CaSR expression on factors of intrinsic neuronal excitability, and examine the relative contributions of $[Ca^{2+}]_o$-regulated changes on VGSC and NALCN gating. In recordings from neocortical neurons, isolated by pharmacological block of excitatory and inhibitory inputs, we determine that neuronal firing is increased by decreasing external divalent concentrations and that this is almost entirely attributable to $[Ca^{2+}]_o$-dependent shifts in VGSC gating. Surprisingly, CaSR deletion substantially shifted VGSC gating, but had no effect on NALCN sensitivity to $[Ca^{2+}]_o$. Taken together our experiments indicate that acute $[Ca^{2+}]_o$-dependent increases in neuronal excitability result from changes in VGSC and NALCN gating and that CaSR contributes by an, as yet, uncharacterized action on VGSCs.

## Results

### CaSR and divalent-dependent neuronal excitability

Increased excitability following the reduction of $[Ca^{2+}]_o$ ($[Ca^{2+}]_o$-dependent excitability) was eliminated by deletion of UNC79 or NALCN in neurons, challenging the long-standing hypothesis that local or diffuse surface charge screening of VGSCs mediated these effects (*Lu et al., 2010*). But how were changes in external divalent ion concentrations transduced to UNC79 and NALCN? We tested if CaSR provided the link, by comparing excitability in wild-type (WT) and nestin Cre-recombinase expressing CaSR null-mutant ($Nes^{Cre}Casr^{fl/fl}$ abbreviated as $Casr^{-/-}$) neurons that were genotyped by PCR (see Materials and methods; *Chang et al., 2008*). Quantification by RT-qPCR indicated >98% reduction in the *Casr* expression levels in neocortical cultures produced from $Casr^{-/-}$ mice compared to Cre-positive WT ($Nes^{Cre}$; *Figure 1—figure supplement 1*). Current clamp recordings were performed to measure the intrinsic, spontaneous action potential firing rate from cultured, neocortical neurons. The cells were also pharmacologically isolated to prevent the confounding influence on action potential firing of changes in synaptic transmission following alterations of $[Ca^{2+}]_o$ and $[Mg^{2+}]_o$ (glutamatergic and GABAergic activity blocked by 10 µM CNQX, 50 µM APV, and 10 µM Gabazine). After establishing the whole-cell configuration, we measured the spontaneous action potential firing rates of conventional WT (conWT), $Nes^{Cre}$, and $Casr^{-/-}$ neurons in physiological Tyrode solution ($T_{1.1}$; containing 1.1 mM) at the resting membrane potential (RMP) and then in reduced $Ca^{2+}$- and $Mg^{2+}$-containing Tyrode ($T_{0.2}$; containing 0.2 mM $[Ca^{2+}]$ and $[Mg^{2+}]$). CaSR and

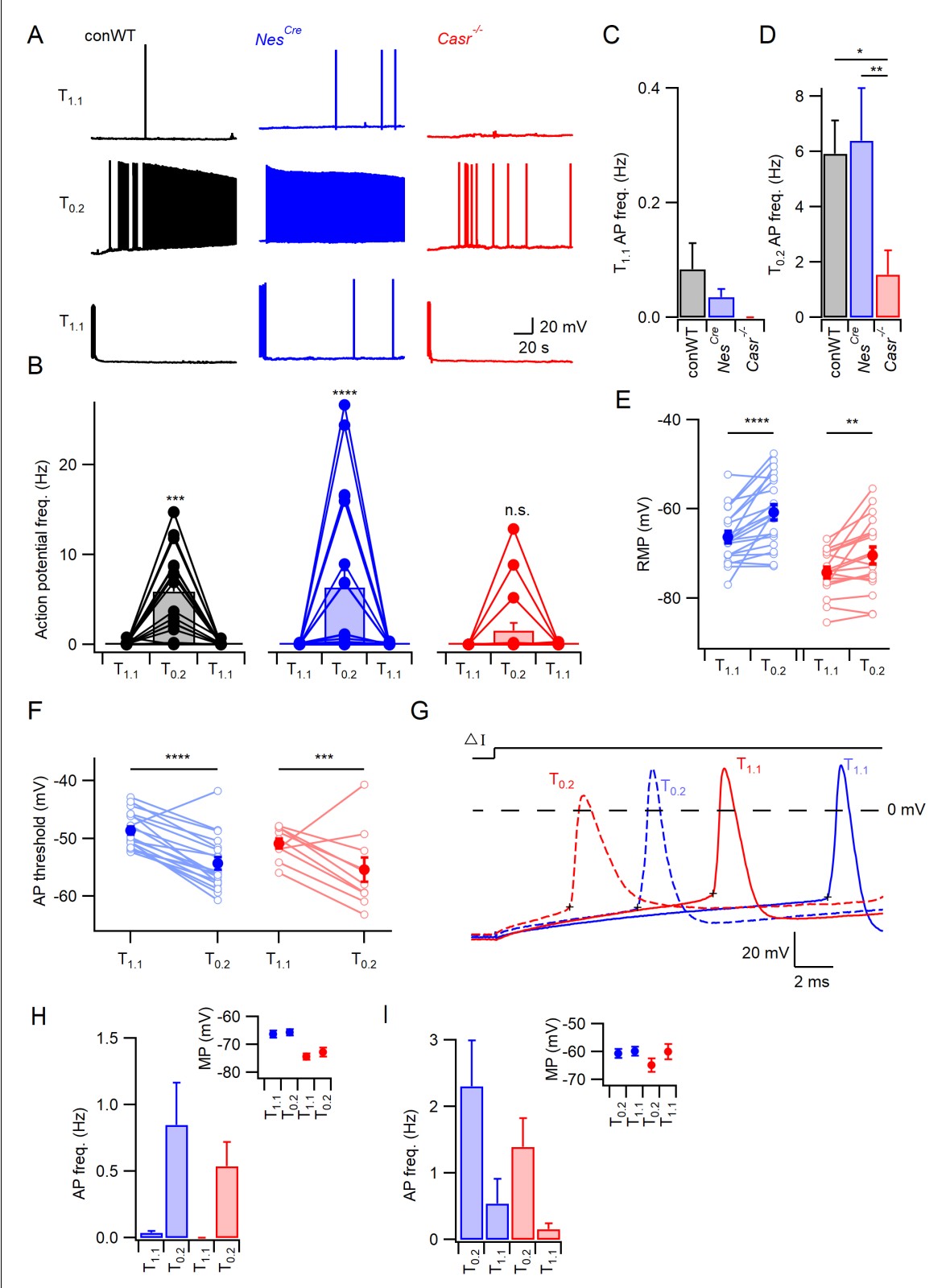

**Figure 1.** CaSR deletion reduces divalent-dependent excitability. (**A**) Spontaneous voltage traces at RMP following the application of solutions with different divalent concentrations ($T_{1.1}$ (upper traces), $T_{0.2}$ (middle), and $T_{1.1}$ recovery (lower)) recorded in three individual neurons with or without CaSR (conWT (black), $Nes^{Cre}$ (blue) and $Casr^{-/-}$ (red)). Each trace depicts 150 s of continuous acquisition. (**B**) Histograms of average action potential (AP) frequency (Hz) recorded using the same solutions: $T_{1.1}$, $T_{0.2}$, and $T_{1.1}$ recovery. Individual recordings represented by open circles linked with lines and

*Figure 1 continued on next page*

*Figure 1 continued*

average is represented with a bar. From left to right graphs depict conWT (n = 18), $Nes^{Cre}$ (n = 21), and $Casr^{-/-}$ (n = 18). ANOVA: Post-hoc tests (Sidak compensated for multiple comparisons here and in all later figures) showed that action potential frequency increased in conWT (p=0.0009) and $Nes^{Cre}$ (P, 0.0001), but not $Casr^{-/-}$ (p=0.6697) neurons when changing from $T_{1.1}$ to $T_{0.2}$ (*Figure 1—source data 1*). (C) Baseline average action potential frequency in $T_{1.1}$. was unaffected by genotype (p>0.999). (D) Average action potential frequency with $T_{0.2}$ application was the same in conWT and $Nes^{Cre}$ (p=0.9831) and higher than in $Casr^{-/-}$ neurons (p=0.013 and 0.0033, respectively). (E) Plot of effect of external divalent concentration and CaSR on RMP. Two-way RM ANOVA indicates that increasing $[Ca^{2+}]_o$ (F (1, 37)=31.65, p<0.0001) and CaSR deletion (F (1, 37)=19.1, p<0.0001) hyperpolarized the RMP without an interaction (F (1, 37)=1.035, p=0.3155). Post-hoc testing indicated RMP was depolarized with the switch to $T_{0.2}$ in both $Nes^{Cre}$ and $Casr^{-/-}$ neurons (p<0.0001 and p=0.0066 for 21 and 18 recordings respectively; *Figure 1—source data 1*). (F) Plot of average action potential threshold in $T_{1.1}$ and $T_{0.2}$ in $Nes^{Cre}$ and $Casr^{-/-}$ neurons elicited as per panel G. Two-way RM ANOVA indicates that reducing $[Ca^{2+}]_o$ hyperpolarized the action potential threshold (F (1, 27)=56.48, p<0.0001) but that genotype had no effect (F (1, 27)=2.284, p=0.1424). $[Ca^{2+}]_o$ was highly effective in both $Nes^{Cre}$ and $Casr^{-/-}$ neurons (p<0.0001 and p=0.0003 for 19 and 10 recordings, respectively). Individual neuron values are represented by open circles linked by lines and averages by filled circles. (G) Exemplar action potentials elicited by current injection in a $Nes^{Cre}$ (blue) and a $Casr^{-/-}$ neuon (red) in $T_{1.1}$ (unbroken) and $T_{0.2}$ (broken). Action potential threshold is indicated by +for the first action potential elicited by current injection (50–200 pA) under the same conditions as panel E. (H) Histogram summarizing effects of divalents on action potential frequency in $Nes^{Cre}$ and Casr-/- neurons after a current injection to counter divalent-dependent depolarization following $T_{0.2}$ application. Two-way RM ANOVA indicates that reducing $[Ca^{2+}]_o$ increases the action potential frequency (F (1, 35)=11.54, p=0.0017) and that this is significant in the $Nes^{Cre}$ but not $Casr^{-/-}$ neurons (p=0.0075 and 0.1555 for 21 and 16 recordings, respectively). Inset shows average membrane potential after the current injection. (I) Histogram summarizing effects of divalents on action potential frequency in $Nes^{Cre}$ and Casr-/- neurons after current injection in $T_{1.1}$ to depolarize membrane potential to value recorded in $T_{0.2}$. Two-way RM ANOVA indicates that reducing $[Ca^{2+}]_o$ increases the action potential frequency (F (1, 35)=45.09, p=0.0004) and that this is significant in the $Nes^{Cre}$ but not $Casr^{-/-}$ neurons (p=0.0044 and 0.056 for 21 and 16 recordings, respectively). Inset shows average membrane potential after the current injection. The online version of this article includes the following source data and figure supplement(s) for figure 1:

**Source data 1.** Action potential frequency and resting membrane potential in conventional WT, $Nes^{Cre}$ and $Casr^{-/-}$ neurons in $T_{1.1}$ or $T_{0.2}$ with no current injection.
**Figure supplement 1.** Casr expression levels reduced in $Casr^{-/-}$ neurons.
**Figure supplement 2.** CaSR deletion reduces divalent-dependent excitability following the generation of action potentials elicited by current injections.

sodium conductance gating are both sensitive to $Ca^{2+}$ and $Mg^{2+}$, with $Ca^{2+}$ being two to three times more potent in both processes (*Frankenhaeuser and Hodgkin, 1957*; *Brown et al., 1993*). Consequently, we modified the concentrations of both divalents to utilize a greater fraction of the dynamic range of the phenomenon under study. The reduction in $[Ca^{2+}]_o$ and $[Mg^{2+}]_o$ caused an increase in spontaneous action potential firing in both types of WT (conventional and $Nes^{Cre}$) neurons within 15 s of the solution change that was substantially attenuated in the $Casr^{-/-}$ neuron (*Figure 1A*, middle row). This divalent-dependent neuronal excitability was reversed within 10 s by changing the bath solution back to physiological external divalent concentrations (*Figure 1A*, lower row). The pooled data from repeat experiments indicated that on average the conWT and $Nes^{Cre}$ neurons were equally sensitive to decreased extracellular divalent concentration and had similarly low spontaneous basal levels of activity (<0.1 Hz, *Figure 1B–D*). Two-way repeated measures (RM) ANOVA confirmed a significant interaction indicating the response to changes of external divalent concentration were dependent on genotype (F (2,54)=3.193, p=0.049, *Table 1*). Post-hoc tests confirmed that the reduction in $[Ca^{2+}]_o$ and $[Mg^{2+}]_o$ substantially increased action potential frequency in conWT and $Nes^{Cre}$ but not $Casr^{-/-}$ neurons (Sidak compensated for multiple comparisons here and in all later tests, *Figure 1B*, p=0.0009,<0.0001, and = 0.6697, respectively). Having confirmed that the conWT and $Nes^{Cre}$ neurons responded quantitatively the same to decreases in external divalents we used $Nes^{Cre}$ neurons alone as controls in subsequent experiments examining CaSR function. These data

**Table 1.** Action potential frequency.

| ANOVA table | SS | DF | MS | F (DFn, DFd) | P value |
|---|---|---|---|---|---|
| Interaction | 130.0 | 2 | 64.99 | F (2, 54)=3.193 | p=0.0489 |
| $[Ca^{2+}]_o$ on AP count | 594.4 | 1 | 594.4 | F (1, 54)=29.21 | p<0.0001 |
| Genotype | 136.1 | 2 | 68.04 | F (2, 54)=3.368 | p=0.0418 |
| Subjects (matching) | 1091 | 54 | 20.20 | F (54, 54)=0.9925 | p=0.5110 |
| Residual | 1099 | 54 | 20.35 | | |

**Table 2.** RMP.

| ANOVA table | SS | DF | MS | F (DFn, DFd) | P value |
|---|---|---|---|---|---|
| Interaction | 14.36 | 1 | 14.36 | F (1, 37)=1.035 | p=0.3155 |
| $[Ca^{2+}]_o$ on RMP | 438.9 | 1 | 438.9 | F (1, 37)=31.65 | p<0.0001 |
| Genotype | 1513 | 1 | 1513 | F (1, 37)=19.10 | p<0.0001 |
| Subjects (matching) | 2930 | 37 | 79.19 | F (37, 37)=5.710 | p<0.0001 |
| Residual | 513.2 | 37 | 13.87 | | |

indicate that CaSR deletion substantially attenuates the increase in spontaneous firing at the RMP produced by reductions in external divalent concentrations in neurons.

## Does CaSR modulate RMP and divalent-dependent depolarization?

If CaSR-mediated NALCN-dependent depolarization is sufficient to account for the response to external divalent reduction, then $Nes^{Cre}$, but not $Casr^{-/-}$, neurons should depolarize in response to the switch to $T_{0.2}$. However, the presence of CaSR and external divalent concentrations were both significant determinants of RMP (zero current injection; two-way RM ANOVA, **Table 2**, F (1,37) =19.1, p<0.0001 and F (1,37)=31.65, p<0.0001, respectively). In fact, the RMP of $Nes^{Cre}$ and $Casr^{-/-}$ neurons both depolarized similarly (**Figure 1E**; 5.6 ± 1.1 mV, p<0.0001 and 3.9 ± 1.2 mV, p=0.0066 respectively) when $T_{0.2}$ was applied indicating the existence of a divalent-sensitive pathway in $Casr^{-/-}$ neurons.

## Divalent-dependent firing persists after hyperpolarization

If NALCN-dependent depolarization is entirely responsible for the extracellular divalent-sensitive changes in neuronal excitability then reversal of this depolarization should prevent (or block) the increase in excitability seen in $T_{0.2}$. To test this prediction, we measured spontaneous action potential frequency in $T_{0.2}$ after adjusting the membrane potential to match the RMP observed in $T_{1.1}$ (current injected to match the membrane potential was unique for each neuron). Action potential frequency in $T_{0.2}$ was reduced by the hyperpolarization, but neurons remained sensitive to reduced divalent concentrations, although not CaSR deletion, indicating mechanisms besides NALCN were involved (**Figure 1H**, **Table 3**; F (1,35)=11.54, p=0.0017, 2-way RM ANOVA). Similarly, in the reciprocal experiment in which the membrane potential in $T_{1.1}$ was depolarized to match that measured at low divalent concentration, the decrease in external divalent concentration increased action potential frequency (**Figure 1I**, **Table 4**; F (1,35)=15.17, p=0.0004, two-way RM ANOVA), and this was significant in $Nes^{Cre}$ but not $Casr^{-/-}$ neurons (**Figure 1I**, p=0.004). Ineffective matching of the membrane potential following solution changes did not account for the persistence of divalent-dependent excitability (insets, **Figure 1H,I**). The sustained sensitivity of spontaneous firing to reduced external divalent concentrations following hyperpolarization of the membrane potential indicated another mechanism, other than NALCN-mediated depolarization, was contributing to the extracellular divalent-sensitive changes in neuronal excitability. Divalent-dependent excitability was also evident in response to transient depolarizing currents (300 ms), with $T_{0.2}$ increasing action potential count over a range of current injections in $Nes^{Cre}$, and to a lesser degree in $Casr^{-/-}$ neurons (**Figure 1—figure supplement 2**). This was observed despite hyperpolarization of the neuron while

**Table 3.** Action potential frequency.

| ANOVA table | SS | DF | MS | F (DFn, DFd) | p Value |
|---|---|---|---|---|---|
| Interaction | 0.3407 | 1 | 0.3407 | F (1, 35)=0.4758 | p=0.4949 |
| $[Ca^{2+}]_o$ at hyperpolarizing injection | 8.262 | 1 | 8.262 | F (1, 35)=11.54 | p=0.0017 |
| Genotype | 0.5380 | 1 | 0.5380 | F (1, 35)=0.7309 | p=0.3984 |
| Subjects (matching) | 25.76 | 35 | 0.7360 | F (35, 35)=1.028 | p=0.4679 |
| Residual | 25.06 | 35 | 0.7161 | | |

**Table 4.** Action potential frequency.

| ANOVA table | SS | DF | MS | F (DFn, DFd) | p Value |
|---|---|---|---|---|---|
| Interaction | 0.6090 | 1 | 0.6090 | F (1, 35)=0.2048 | p=0.6536 |
| $[Ca^{2+}]_o$ at depolarizing injection | 45.09 | 1 | 45.09 | F (1, 35)=15.17 | p=0.0004 |
| Genotype | 5.982 | 1 | 5.982 | F (1, 35)=0.9959 | p=0.3252 |
| Subjects (matching) | 210.2 | 35 | 6.006 | F (35, 35)=2.020 | p=0.0204 |
| Residual | 104.1 | 35 | 2.973 | | |

in $T_{0.2}$ to the resting membrane potential measured in $T_{1.1}$, consistent with it occurring independent of any NALCN-mediated depolarization.

The action potential threshold was measured to determine if there was a difference in the apparent excitability of $Nes^{Cre}$ and $Casr^{-/-}$ neurons. Action potentials were elicited in $T_{1.1}$ and $T_{0.2}$ using minimal current injection (50–250 pA) and the threshold measured as the point at which dV/dt reached 20 mV/ms (*Figure 1G*, membrane potential-corrected as in *Figure 1H* to minimize the effect of the depolarization itself). The action potential threshold was hyperpolarized from −48.6 ± 0.7 mV to −54.3 ± 1.1 mV with the switch from $T_{1.1}$ to $T_{0.2}$ in $Nes^{Cre}$ neurons (*Figure 1F*) which would have increased excitability. However, the same effect was observed in $Casr^{-/-}$ neurons (−50.9 ± 0.86 mV to −55.4 ± 2.1 mV; F (1,27)=56.48, p<0.0001, two-way RM ANOVA, *Table 5*). As CaSR deletion did not affect action potential threshold under these conditions (*Figure 1F*), spike generation presumably occurred more frequently in the $Nes^{Cre}$ neurons due to the relatively depolarized membrane potential (8 mV positive than $Casr^{-/-}$ neurons, *Figure 1E*). The lack of effect of CaSR on spike threshold (F (1, 27)=2.284, p=0.142) in these experiments, indicated the reduced divalent sensitivity of $Casr^{-/-}$ (*Figure 1E,F*) was not simply due to altered action potential threshold.

Overall these data support the idea that CaSR played a role in mediating divalent dependent changes in excitability, but that neurons also possessed CaSR-independent mechanisms to fully account for the divalent-dependent excitability.

## CaSR effects on divalent-dependent excitability attenuated by matching membrane potential

Further mechanistic complexity was suggested by the effects of CaSR and $[Ca^{2+}]_o$ on RMP. This lead to a number of additional questions including: does the difference in RMP contribute to the difference in divalent-dependent excitability between $Nes^{Cre}$ and $Casr^{-/-}$ neurons, how do decreases in $[Ca^{2+}]_o$ depolarize $Casr^{-/-}$ neurons, and is this pathway present in $Nes^{Cre}$ neurons? To address the first of these questions, we compared the response of $Nes^{Cre}$ and $Casr^{-/-}$ neurons to changes in extracellular divalent concentrations after removing the confounding variation in RMP. After establishing a stable current-clamp recording in $T_{1.1}$ we injected a standing current ($I_a$) until the resting membrane potential was −70 mV. We then recorded for 50 s before switching the bath solution to $T_{0.2}$. As before, there was a small depolarization followed by an increase in action potential frequency in $Nes^{Cre}$ neurons (*Figure 2A,B*). To test if this increase in excitability was fully attributable to divalent-dependent depolarization we adjusted the standing current ($I_b$) until the membrane potential was −70 mV and then measured the action potential frequency (*Figure 2C*). In the exemplar, action potential firing was reduced by the hyperpolarization but remained higher in $T_{0.2}$ at −70

**Table 5.** Action potential threshold.

| ANOVA table | SS | DF | MS | F (DFn, DFd) | p Value |
|---|---|---|---|---|---|
| Interaction | 0.5225 | 1 | 0.5225 | F (1, 27)=0.07478 | p=0.7866 |
| $[Ca^{2+}]_o$ on AP threshold | 394.6 | 1 | 394.6 | F (1, 27)=56.48 | p<0.0001 |
| Genotype | 54.34 | 1 | 54.34 | F (1, 27)=2.284 | p=0.1424 |
| Subjects (matching) | 642.5 | 27 | 23.80 | F (27, 27)=3.406 | p=0.0011 |
| Residual | 188.6 | 27 | 6.987 | | |

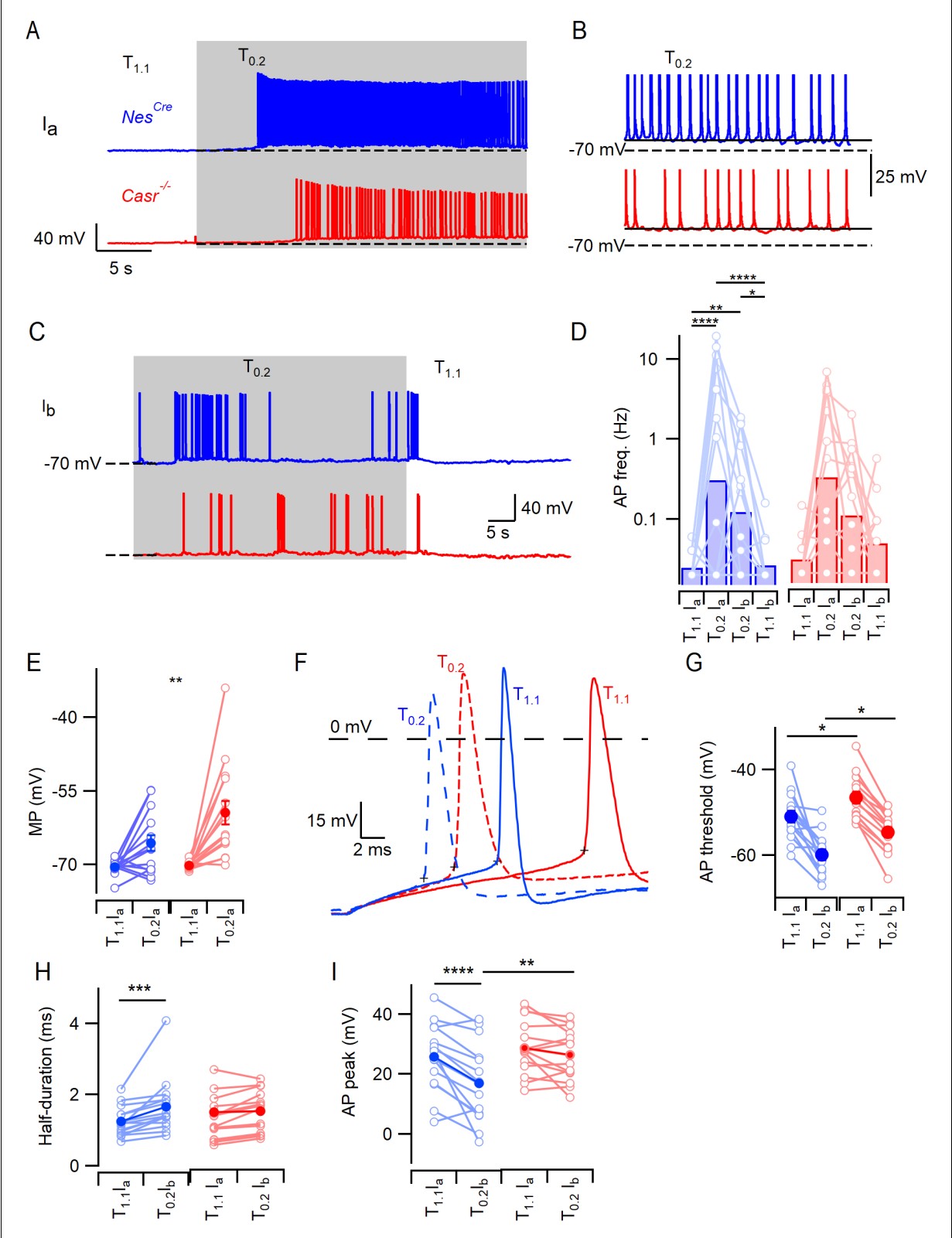

**Figure 2.** CaSR deletion does not affect divalent-dependent excitability at equivalent membrane potential. (**A**) Exemplary traces showing the divalent-dependent increase in neuronal excitability following the switch from $T_{1.1}$ to $T_{0.2}$ (change indicated by upper trace) in $Nes^{Cre}$ (blue) and $Casr^{-/-}$ (red) neurons when initial membrane potentials matched at −70 mV (broken line). (**B**) Expanded view of the final 5 s of traces in A illustrating sustained depolarization from following $T_{0.2}$ application. (**C**) Exemplary traces showing the divalent-dependent decrease in neuronal excitability following the

*Figure 2 continued on next page*

*Figure 2 continued*

switch from $T_{0.2}$ to $T_{1.1}$ (change indicated by upper trace) in $Nes^{Cre}$ (blue) and $Casr^{-/-}$ (red) neurons when initial membrane potentials matched at −70 mV. Same recordings as A. (**D**) Histogram of average divalent-dependent changes in action potential frequency (Hz) in $Nes^{Cre}$ (blue) and $Casr^{-/-}$ (red) neurons when initial voltage is −70 mV in $T_{1.1}$ (Ia) or $T_{0.2}$ (Ib). Two-way RM ANOVA performed after logarithmic transformation indicates that reducing $[Ca^{2+}]_o$ increases the action potential frequency (F (3, 87)=17.97, p<0.0001) similarly in $Nes^{Cre}$ and $Casr^{-/-}$ neurons (F (1, 29)=0.2005, p=0.6577; *Figure 2—source data 1*). Post-hoc tests indicate significant differences between action potential frequency in $T_{1.1}$ and $T_{0.2}$ regardless of the holding current but not between action potential frequency recorded at different holding currents and the same solutions (Ia or Ib; *Table 7*). (**E**) Membrane potential depolarization following the switch to $T_{0.2}$ from $T_{1.1}$. Two-way RM ANOVA indicates that reducing $[Ca^{2+}]_o$ (F (1, 29)=29.22, p<0.0001) and CaSR deletion (F (1, 29)=4.874, p=0.0353) significantly depolarized the membrane potential but that there was no interaction (F (1, 29)=4.055, p=0.0534). Post-hoc testing indicate that membrane potentials were matched using current injection in $T_{1.1}$ (-70.5 ± 0.4 mV and −70.2 ± 0.2 mV for $Nes^{Cre}$ and $Casr^{-/-}$ neurons respectively, p=0.985) but different in $T_{0.2}$ (-65.6 ± 1.6 mV and –59.4 ± 2.4 mV, p=0.0083). (**F**) Exemplar action potentials elicited by current injection from −70 mV in a $Nes^{Cre}$ (blue) and a $Casr^{-/-}$ neuron (red) in solutions $T_{1.1}$ (unbroken) and $T_{0.2}$ (broken). Action potential threshold is indicated by +symbol for the first action potential elicited by current injection (50 to 200 pA). (**G**) Plot of average action potential threshold in $T_{1.1}$ and $T_{0.2}$ in $Nes^{Cre}$ and $Casr^{-/-}$ neurons, elicited as per panel F here and in subsequent panels. Two-way RM ANOVA indicates that reducing $[Ca^{2+}]_o$ hyperpolarized the action potential threshold (F (1, 25)=51.66, p<0.0001), whereas CaSR deletion had the opposite effect (F (1, 25)=10.52, p=0.0033). There was no interaction (*Table 9A*). Post-hoc tests indicate that the action potential thresholds in solutions $T_{1.1}$ and $T_{0.2}$ were depolarized similarly by CaSR deletion (5.3 ± 2.0 mV and 5.5 ± 2.0 mV, p=0.020 and 0.017) in $Nes^{Cre}$ and $Casr^{-/-}$ neurons, respectively. (**H**) Plot of average action potential half-duration in $T_{1.1}$ and $T_{0.2}$ in $Nes^{Cre}$ and $Casr^{-/-}$ neurons. Two-way RM ANOVA indicates that reducing $[Ca^{2+}]_o$ prolonged the action potential half-duration (F (1, 28) =19.73, p=0.0001). (**I**) Plot of average action potential peak in $T_{1.1}$ and $T_{0.2}$ in $Nes^{Cre}$ and $Casr^{-/-}$ neurons. The action potential peaks were higher in $T_{1.1}$ and in Casr-/- neurons (*Table 9C*).

The online version of this article includes the following source data for figure 2:

**Source data 1.** Action potential frequency in $Nes^{Cre}$ and $Casr^{-/-}$ neurons in $T_{1.1}$ or $T_{0.2}$ with standing currents $I_a$ and $I_b$.

mV than in $T_{1.1}$ at −70 mV (*Figure 2A–C*) confirming CaSR-mediated depolarization was not acting alone to increase the excitability. The $Casr^{-/-}$ neurons responded similarly to $T_{0.2}$ and hyperpolarization (*Figure 2A–C*) indicating the effect was not mediated by CaSR. We compared the average effects of $T_{1.1}$ at −70 mV with $I_a$, $T_{0.2}$ with $I_a$, and $T_{0.2}$ at −70 mV with $I_b$ on $Nes^{Cre}$ and $Casr^{-/-}$ genotypes (*Figure 2D*, *Table 6*) using a 2-way RM ANOVA. Extracellular divalent concentration and current injection substantially affected action potential frequency (F (3, 87)=17.97, p<0.0001). CaSR deletion did not impact the response to extracellular divalent concentration when $Nes^{Cre}$ and $Casr^{-/-}$ neuron recordings were started at a membrane potential of −70 mV l (F (1, 29)=0.2005, p=0.6577). Post-hoc testing showed that excitability was increased in $T_{0.2}$ compared with $T_{1.1}$ regardless which of the two holding currents were used (*Figure 2D*; *Table 7*). After injection of $I_a$ to set the membrane potential to −70 mV, the switch from $T_{1.1}$ to $T_{0.2}$ still significantly depolarized the membrane potential (*Figure 2E*; *Table 8*, Two-way RM ANOVA, F (1, 29)=29.22, p<0.0001) as did CaSR deletion (F (1, 29)=4.874, p=0.0353). Post-hoc testing indicate that the membrane potential in $T_{0.2}$ was more depolarized in the $Casr^{-/-}$ than in $Nes^{Cre}$ neurons (*Figure 2B,E*; −65.6 ± 1.6 mV vs −59.4 ± 2.4 mV, p=0.0083). Taken together, these experiments indicate CaSR-NALCN signaling was not contributing to the difference in divalent-dependent excitability between $Nes^{Cre}$ and $Casr^{-/-}$ neurons but that these differences may be due to genotype-dependent differences in RMP or intrinsic excitability.

**Table 6.** Action potential frequency.

| ANOVA table | SS | DF | MS | F (DFn, DFd) | p Value |
|---|---|---|---|---|---|
| Interaction | 0.4305 | 3 | 0.1435 | F (3, 87)=0.3481 | p=0.7906 |
| $[Ca^{2+}]_o$ and I | 22.23 | 3 | 7.410 | F (3, 87)=17.97 | p<0.0001 |
| Genotype | 0.1341 | 1 | 0.1341 | F (1, 29)=0.2005 | p=0.6577 |
| Subjects (matching) | 19.41 | 29 | 0.6692 | F (29, 87)=1.623 | p=0.0445 |
| Residual | 35.87 | 87 | 0.4123 | | |

**Table 7.** Action potential frequency.

| Sidak's multiple comparisons test | Mean diff. | 95% CI of diff. | Significant? | Summary | Adjusted p value |
|---|---|---|---|---|---|
| $T_{1.1}$ Ia vs. $T_{0.2}$ Ia | −1.059 | −1.498 to −0.6200 | Yes | **** | <0.0001 |
| $T_{1.1}$ Ia vs. $T_{0.2}$ Ib | −0.6203 | −1.059 to −0.1813 | Yes | ** | 0.0016 |
| $T_{1.1}$ Ia vs. $T_{1.1}$ Ib | −0.1163 | −0.5554 to 0.3227 | No | ns | 0.9797 |
| $T_{0.2}$ Ia vs. $T_{0.2}$ Ib | 0.4387 | −0.0003709 to 0.8778 | No | ns | 0.0503 |
| $T_{0.2}$ Ia vs. $T_{1.1}$ Ib | 0.9427 | 0.5037 to 1.382 | Yes | **** | <0.0001 |
| $T_{0.2}$ Ib vs. $T_{1.1}$ Ib | 0.5040 | 0.06496 to 0.9431 | Yes | * | 0.0160 |

## Voltage-gated sodium channels contribute to divalent-dependent excitability

Reversal of the divalent-dependent depolarization did not completely block the increased excitability associated with the switch to $T_{0.2}$ (*Figures 1E, F* and *2D*) indicating another mechanism other than NALCN was responsible. We tested if voltage-gated channels were contributing by to divalent-dependent excitability by examining action potential threshold in neurons held at a membrane potential of −70 mV. Action potential threshold was hyperpolarized by 8 mV on average following the change from $T_{1.1}$ to $T_{0.2}$ in *Nes$^{Cre}$* and *Casr$^{-/-}$* neurons (*Figure 2F,G*, *Table 9A*; $F_{(1, 25)}=51.66$, p<0.0001). Furthermore, the action potential threshold was relatively depolarized in the *Casr$^{-/-}$* neurons in $T_{1.1}$ and $T_{0.2}$ ($5.3 \pm 2.0$ mV (p=0.020) and $5.5 \pm 2.0$ mV (p=0.017) respectively), indicating *Nes$^{Cre}$* neurons possessed increased excitability and increased sensitivity to decreases in external divalent concentration (*Figure 2F,G*). The action potential half-width recorded under the same conditions, was also sensitive to the reduction of divalent concentration but unaffected by CaSR deletion (*Figure 2H,I Table 9B*). ANOVA indicated that the switch to $T_{0.2}$ from $T_{1.1}$ broadened action potential half-width ($F_{(1,28)}=19.7$, p=0.0001). The genotype and $[Ca^{2+}]_o$ interacted to both affect action potential peak voltage (*Figure 2I*, *Table 9C*; $F_{(1, 28)}=6.76$, p=0.015) with the peak potential being reduced by $T_{0.2}$ in the *Nes$^{Cre}$* (p<0.0001) but not *Casr$^{-/-}$* neurons (p=0.34).

We examined the properties of VGSCs and voltage-gated potassium channels (VGPCs) to determine the reason for the altered action potential threshold. VGSCs were isolated in neocortical neurons and the current-voltage characteristics examined. Families of VGSC currents were activated in neurons after 2–4 weeks in culture. Maximum VGSC currents were elicited at −30 mV and averaged $-8.0 \pm 0.8$ nA (n = 7) and $-8.8 \pm 2.8$ nA (n = 6) in *Nes$^{Cre}$* and *Casr$^{-/-}$* neurons, respectively. The current-voltage curve shifted in a hyperpolarizing direction with the switch from $T_{1.1}$ to $T_{0.2}$ but extensive neuronal processes limited the quality of the voltage-clamp and prevented useful analysis. We examined VGSC gating in nucleated outside-out patches (*Sather et al., 1992*; *Almog et al., 2018*) to ensure better voltage control. VGSC currents were elicited by voltage steps from −80 mV (10 mV increments to 40 mV). In $T_{0.2}$, the VGSC inactivation (see below) resulted in smaller currents that were more sensitive to depolarization (bold traces elicited by steps to −50 mV, *Figure 3A*) as previously observed (*Frankenhaeuser and Hodgkin, 1957*; *Campbell and Hille, 1976*; *Armstrong and Cota, 1991*). Divalent sensitivity was confirmed in the normalized current-voltage plot for both *Nes$^{Cre}$* (blue, n = 8) and *Casr$^{-/-}$* (red, n = 11) neurons (*Figure 3A,C*). VGSC current inactivation was studied using a test pulse to −20 mV, each of which was preceded by a conditioning step (100 ms) to between −140 mV and −20 mV. In $T_{1.1}$ we observed less inactivation than in $T_{0.2}$ (*Figure 3B*, bold

**Table 8.** Membrane potential with Ia.

| ANOVA table | SS | DF | MS | F (DFn, DFd) | p Value |
|---|---|---|---|---|---|
| Interaction | 131.7 | 1 | 131.7 | $F_{(1, 29)}=4.055$ | p=0.0534 |
| $[Ca^{2+}]_o$ | 949.2 | 1 | 949.2 | $F_{(1, 29)}=29.22$ | p<0.0001 |
| Genotype | 162.7 | 1 | 162.7 | $F_{(1, 29)}=4.874$ | p=0.0353 |
| Subjects (matching) | 968.0 | 29 | 33.38 | $F_{(29, 29)}=1.028$ | p=0.4711 |
| Residual | 942.0 | 29 | 32.48 | | |

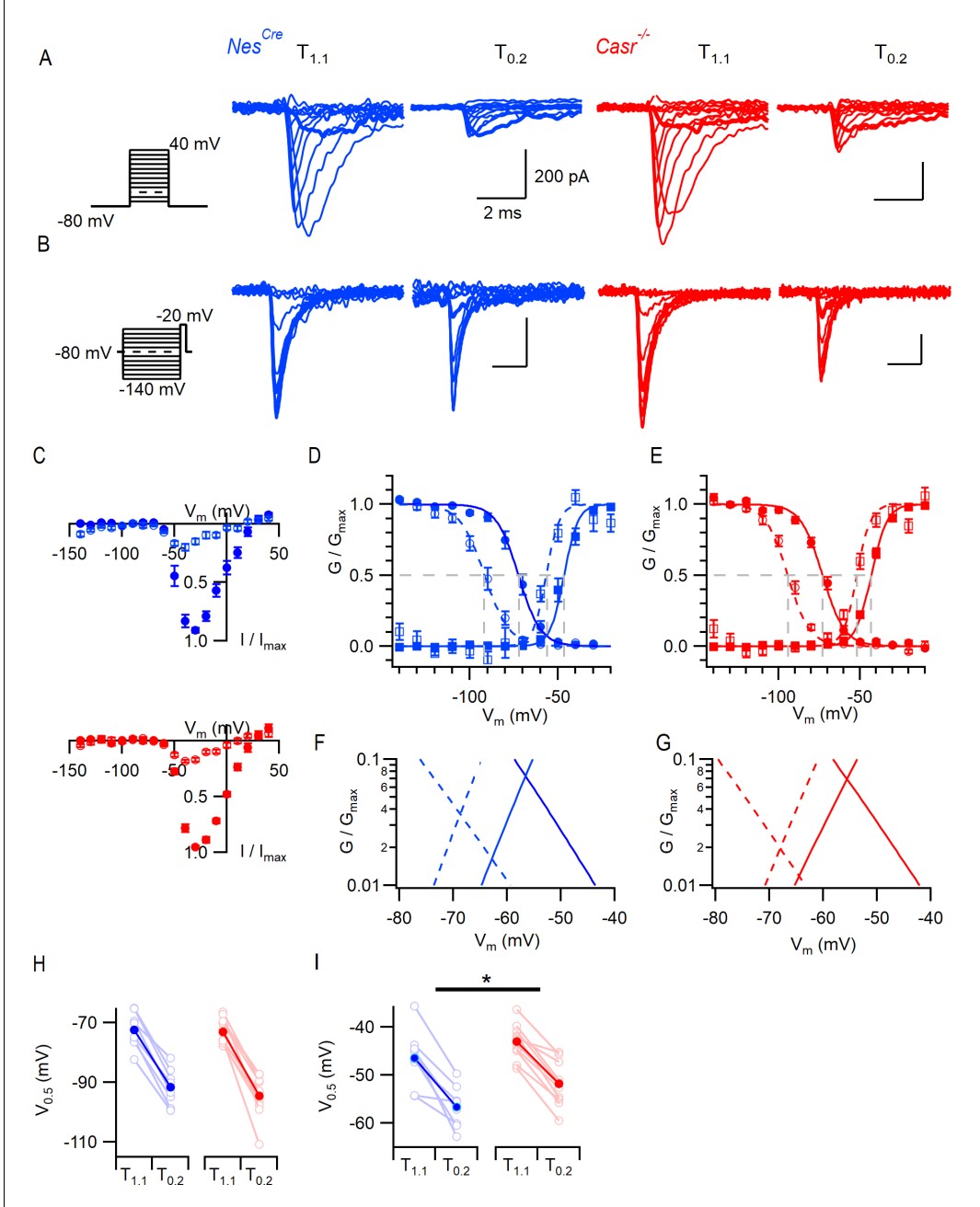

**Figure 3.** CaSR deletion and external divalent concentration affect VGSC current gating. (**A**) Exemplary traces showing VGSC currents activated by voltage steps from −80 in 10 mV increments (left), in nucleated patches isolated from $Nes^{Cre}$ (blue) and $Casr^{-/-}$ (red) neurons in solutions $T_{1.1}$ and $T_{0.2}$. The VGSC currents elicited by 10 ms depolarizations to −50 mV (bold) were greater following the switch to $T_{0.2}$. (**B**) Exemplary traces showing VGSC currents activated by voltage steps to −20 mV following a 100 ms conditioning step (left), in the same patches as (A) using solutions $T_{1.1}$ and $T_{0.2}$. The VGSC currents elicited following conditioning steps to −80 mV (bold) were smaller following the switch to $T_{0.2}$. (**C**) Current-voltage plots of average normalized VGSC currents in nucleated patches from $Nes^{Cre}$ (n = 8) and $Casr^{-/-}$ (n = 11) neurons in $T_{1.1}$ (filled circles) and $T_{0.2}$ (open circles). Currents were normalized using the maximum VGSC current in each recording. (**D**) Plot of average normalized conductance versus voltage in patches from $Nes^{Cre}$ neurons for activation (square, n = 8) and inactivation (circle, n = 8) in solutions $T_{1.1}$ (filled) and $T_{0.2}$ (open). Boltzmann curves are drawn using average values from individual fits and gray broken lines indicate $V_{0.5}$ values for each condition. (**E**) Plot of average normalized conductance versus voltage in patches from $Casr^{-/-}$ neurons for activation (square, n = 11) and inactivation (circle, n = 12) in solutions $T_{1.1}$ (filled) and $T_{0.2}$ (open). Boltzmann curves are drawn using average values from individual fits and gray broken lines indicate $V_{0.5}$ values for each condition. Inset shows plot expanded to emphasize voltage dependence of the window currents. (**F** and **G**) represent the plots of D and E expanded to emphasize the voltage dependence of the window currents. (**H**) Histogram showing $V_{0.5}$ for VGSC inactivation in $T_{1.1}$ and $T_{0.2}$ in patches from $Nes^{Cre}$ and $Casr^{-/-}$ neurons. (**I**) Histogram showing $V_{0.5}$ for VGSC activation in $T_{1.1}$ and $T_{0.2}$ in patches from $Nes^{Cre}$ and $Casr^{-/-}$ neurons.

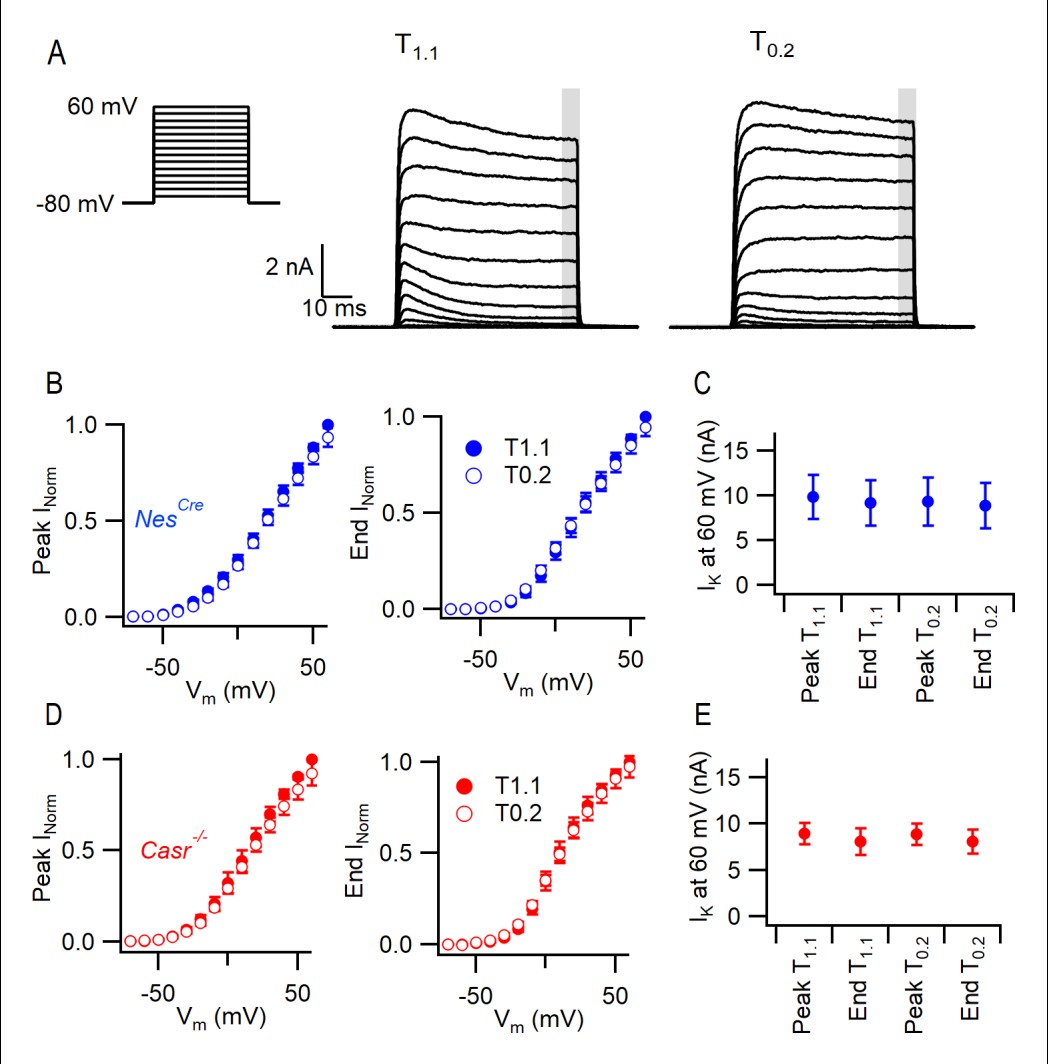

**Figure 4.** CaSR deletion and external divalent concentration do not significantly affect VGPC current gating. (A) Exemplary traces showing VGPC currents activated by voltage steps from $-80$ in 10 mV increments (left), in a $Nes^{Cre}$ neuron in solutions $T_{1.1}$ and $T_{0.2}$. The outward currents elicited by the 50 ms voltage step were measured at peak and at the end of the step (average of last 5 ms indicated by gray bar). (B) Current voltage-plot of average normalized VGPC currents (n = 10) in $Nes^{Cre}$ neurons in $T_{1.1}$ (filled circles) and $T_{0.2}$ (open circles) at peak or end of step. Currents were normalized using the maximum outward current in each condition here and below. (C) Peak and end outward currents at 60 mV elicited in same neurons as B. Two-way RM ANOVA indicates that peak and outward currents were not different in $T_{1.1}$ or $T_{0.2}$ ((3, 57)=1.347), p=0.2683 nor were they affected by CaSR deletion (data from E, (1, 19)=1.231, p=0.2811). (D) Current voltage-plot of average normalized VGPC currents (n = 11) in $Casr^{-/-}$ neurons in $T_{1.1}$ (filled circles) and $T_{0.2}$ (open circles) at peak or end of step. (E) Peak and end outward currents at 60 mV elicited in same neurons as D.

traces show currents elicited following prepulse to $-80$ mV). We compared the effects of $[Ca^{2+}]_o$ and CaSR deletion on VGSC current inactivation using plots of normalized conductance and measuring the half maximal voltage ($V_{0.5}$; circles, *Figure 3D,E*). The reduction in divalent concentration left-shifted $V_{0.5}$ (F (1, 18)=56, p<0.0001, 2-way RM ANOVA, *Table 10*) but CaSR deletion did not (F (1, 18)=0.563, p=0.463). The switch from $T_{1.1}$ to $T_{0.2}$ shifted $V_{0.5}$ by $-20$ and $-21$ mV in $Nes^{Cre}$ and $Casr^{-/-}$, respectively ($-72 \pm 2$ to $-92 \pm 2$ mV and $-73 \pm 1$ to $-94 \pm 2$ mV).

We also tested how the VGSC activation was affected by CaSR and $[Ca^{2+}]_o$. The peak inward VGSC currents (*Figure 3B,D*) were divided by the driving voltage and then plotted as conductance-voltage plots. The normalized conductance plots (squares, *Figure 3E,F*) indicate that the switch

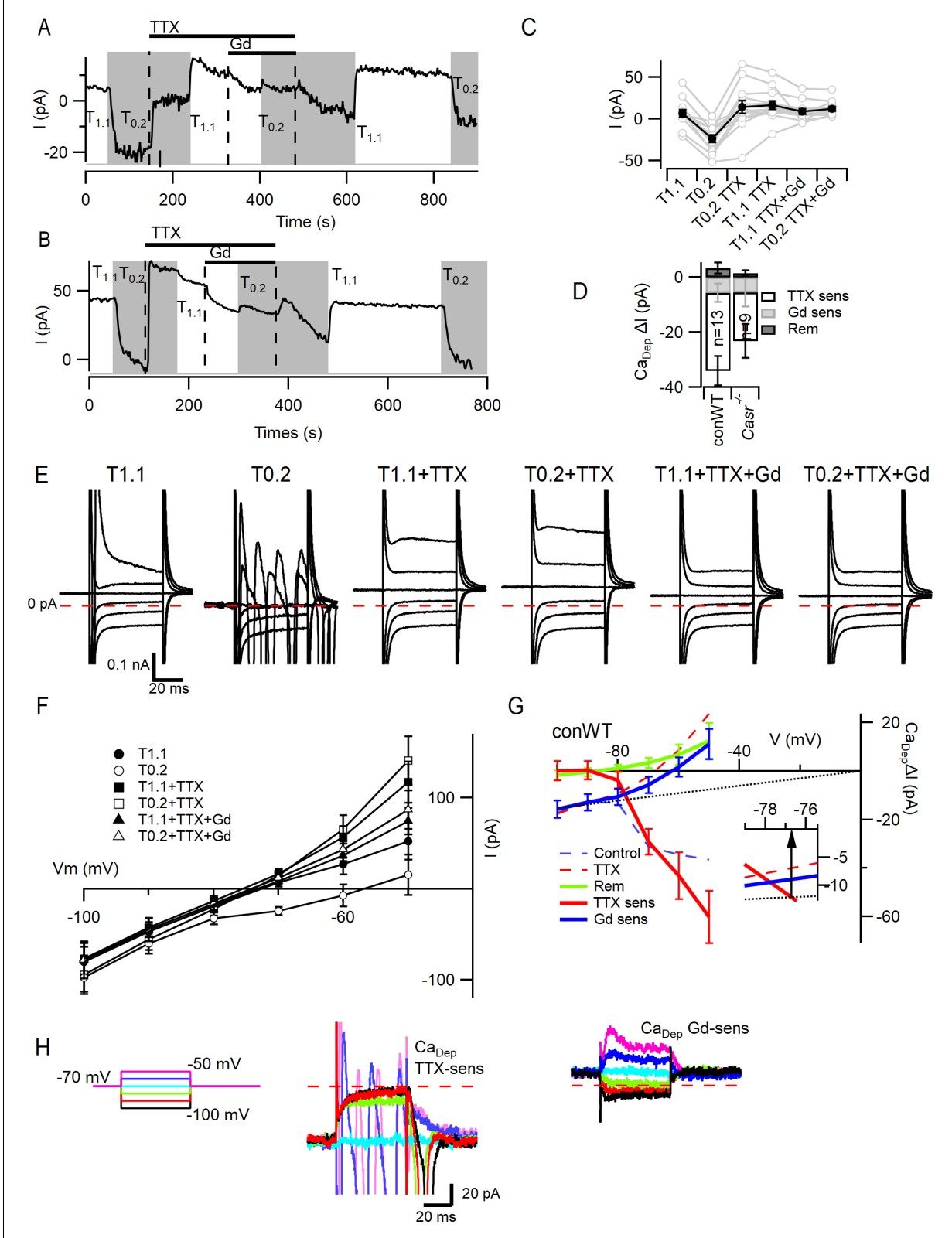

**Figure 5.** VGSC current activation by decreased external divalent concentration. (**A,B**) Plots illustrating the responses of the basal currents in two WT neurons during application of $T_{1.1}$ and $T_{0.2}$ before and during TTX or TTX and $Gd^{3+}$. Average basal currents were measured over 50 ms every 2 s with $T_{1.1}$ and $T_{0.2}$ application indicated by vertical shading (gray represents $T_{0.2}$) and blockers application by horizontal bars and broken vertical lines. (**C**) Plot of average basal current measurements (filled circles) and individual neurons (open circles) in each solution condition in conWT (n = 13) neurons. Each

*Figure 5 continued on next page*

*Figure 5 continued*

basal current represents the average value recorded during last 20 s of the specific solution application. (**D**) Average $[Ca^{2+}]_o$ dependent basal currents sensitive to TTX and $Gd^{3+}$ calculated by subtraction of data in C and the remaining $[Ca^{2+}]_o$ dependent current after application of both blockers for conWT neurons. (**E**) Exemplar traces of currents elicited by 50 ms voltage steps between −100 and −50 mV during application of solutions described in C. (**F**) Plots of the average currents over the last 5 ms of each voltage step in all six solutions for conWT (n = 13). (**G**) Plots of the average $[Ca^{2+}]_o$ dependent currents derived by subtraction of conWT data (**F**) resolved as total or control (broken blue), in the presence of TTX (broken red), and in the presence of TTX and $Gd^{3+}$ (remainder green). The TTX-sensitive (solid red), $Gd^{3+}$-sensitive (solid blue) and NALCN (dotted line) component currents were obtained by further subtraction. Inset shows expanded view at intercept of TTX-sensitive and NALCN components. (**H**) Exemplars of the TTX- and $Gd^{3+}$-sensitive $[Ca^{2+}]_o$-dependent currents. Broken red line represents zero current line.

The online version of this article includes the following figure supplement(s) for figure 5:

**Figure supplement 1.** VGSC current activation by decreased external divalent concentrationin *Casr*$^{-/-}$ neurons.

from $T_{1.1}$ to $T_{0.2}$ significantly facilitated VGSC activation consistent with VGSCs in other excitable cells ($V_{0.5}$ was hyperpolarized by 10 mV; F (1, 17)=98, p<0.0001; two-way RM ANOVA, *Table 11*; *Hille, 2001*). Switching from $T_{1.1}$ to $T_{0.2}$ shifted $V_{0.5}$ by −11 mV and −9 mV in $Nes^{Cre}$ and *Casr*$^{-/-}$ neurons respectively (−46 ± 2 to −57 ± 2 mV and −43 ± 1 to −52 ± 1 mV). The unexpected shift in $V_{0.5}$ for VGSC activation in Casr$^{-/-}$ neurons will reduce the likelihood of VGSC activation (F (1, 17)=4.8, p=0.04) in these cells (*Figure 3I*). Overlap of the inactivation and activation conductance plots represents the voltage range over which persistent VGSC currents, or window currents, are likely to occur (*Chadda et al., 2017*). Divalent reduction hyperpolarized this region of overlap toward the RMP (*Figure 3F,G* insets) increasing the likelihood that persistent VGSC currents were activated at resting membrane potential and therefore contributing to divalent-dependent excitability. The depolarization of VGSC activation gating that resulted from CaSR deletion (*Figure 3I*), shifted the area of conductance curve overlap for $T_{0.2}$ in a depolarizing direction (*Figure 3G*). This effect would reduce the fraction of VGSCs available for activation by $T_{0.2}$ at the more hyperpolarized RMPs and explain the reduced the likelihood of spontaneous action potential generation in Casr$^{-/-}$ neurons (*Figure 1*).

VGPC currents were isolated and recorded in $Nes^{Cre}$ and *Casr*$^{-/-}$ neurons in $T_{1.1}$ and $T_{0.2}$ solutions after blocking contaminating currents. Currents were elicited by a series of 60 ms steps from −70 mV to 60 mV in 10 mV increments (*Figure 4*). The VGPC current amplitudes were measured at the peak and at the end of the depolarizing step (normalized to the value at 60 mV in $T_{1.1}$). Neither the

**Table 9.** Action potential threshold recorded at −70 mV.

| ANOVA table | SS | DF | MS | F (DFn, DFd) | p Value |
|---|---|---|---|---|---|
| Interaction | 0.06658 | 1 | 0.06658 | F (1, 25)=0.004070 | p=0.9496 |
| $[Ca^{2+}]_o$ | 845.1 | 1 | 845.1 | F (1, 25)=51.66 | p<0.0001 |
| Genotype | 391.0 | 1 | 391.0 | F (1, 25)=10.52 | p=0.0033 |
| Subjects (matching) | 929.5 | 25 | 37.18 | F (25, 25)=2.273 | p=0.0225 |
| Residual | 408.9 | 25 | 16.36 | | |
| (B) Action potential threshold recorded at −70 mV | | | | | |
| Interaction | 2.008e-07 | 1 | 2.008e-07 | F (1, 28)=2.800 | p=0.1054 |
| $[Ca^{2+}]_o$ | 1.415e-06 | 1 | 1.415e-06 | F (1, 28)=19.73 | p=0.0001 |
| Genotype | 3.050e-07 | 1 | 3.050e-07 | F (1, 28)=0.4545 | p=0.5057 |
| Subjects (matching) | 1.879e-05 | 28 | 6.710e-07 | F (28, 28)=9.358 | p<0.0001 |
| Residual | 2.008e-06 | 28 | 7.170e-08 | | |
| (C) Action potential threshold recorded at −70 mV | | | | | |
| Interaction | 0.0001602 | 1 | 0.0001602 | F (1, 28)=6.758 | p=0.0147 |
| $[Ca^{2+}]_o$ | 0.0004821 | 1 | 0.0004821 | F (1, 28)=20.34 | p=0.0001 |
| Genotype | 0.001193 | 1 | 0.001193 | F (1, 28)=5.891 | p=0.0219 |
| Subjects (matching) | 0.005669 | 28 | 0.0002025 | F (28, 28)=8.541 | p<0.0001 |
| Residual | 0.0006637 | 28 | 2.370e-005 | | |

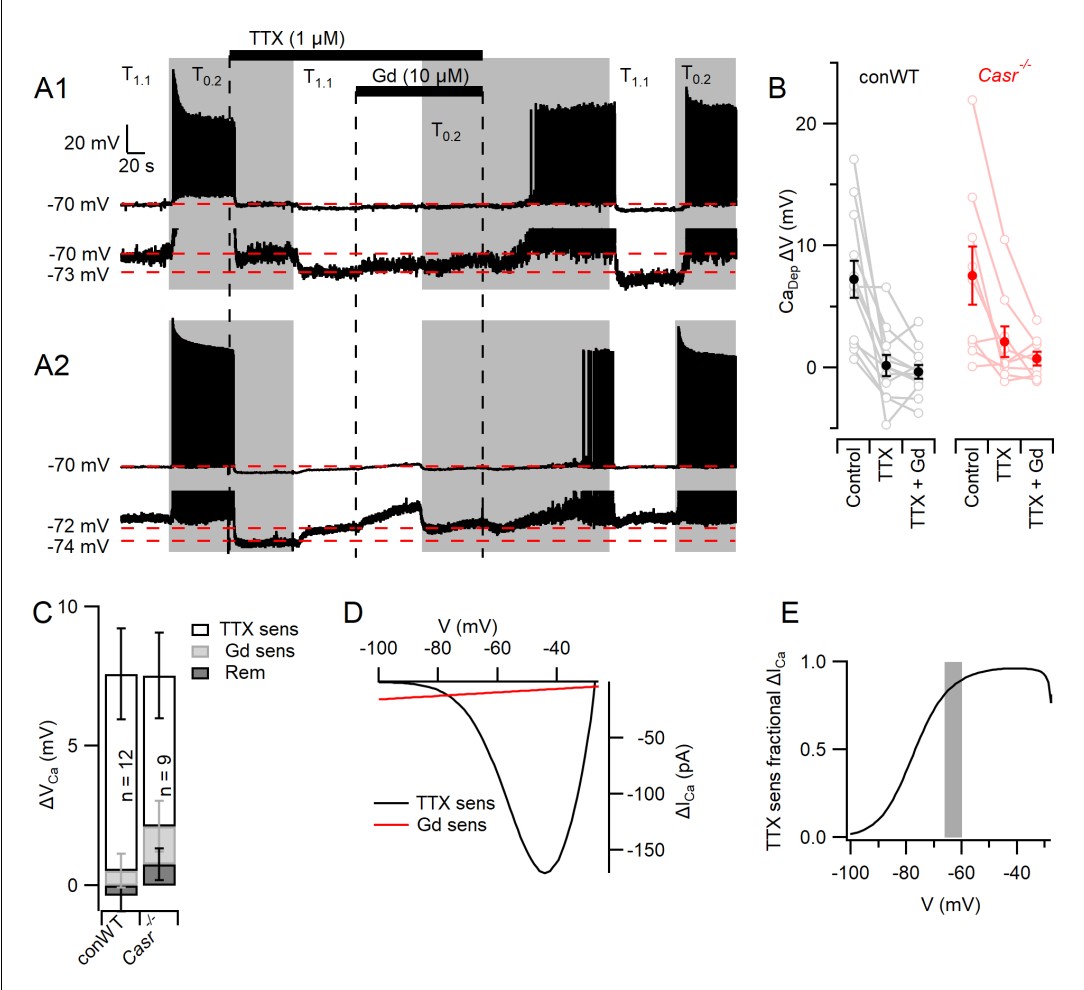

**Figure 6.** Divalent-dependent depolarization is almost entirely mediated via VGSCs. (**A**) The response of the membrane potential in two WT neurons during application of $T_{1.1}$ and $T_{0.2}$ before and during TTX or TTX and $Gd^{3+}$. $T_{1.1}$ and $T_{0.2}$ application is indicated by vertical shading (gray represents $T_{0.2}$) and blocker applications by horizontal bars and broken vertical lines. The broken red line indicates −70 mV. Voltage-expanded view of the trace illustrates that in the presence of TTX, hyperpolarization (A1) and depolarization (A2) may occur following the switch to $T_{1.1}$. Membrane potential values highlighted by broken red lines. (**B**) Plot of average (filled circles) and individual (open circles) $Ca^{2+}$-dependent voltage changes (filled circles) following the switch from $T_{1.1}$ to $T_{0.2}$ (by subtraction of average between-spike membrane potential over the last 10 s of each solution application). Each solution applied to conWT (n = 12) and $Casr^{-/-}$ (n = 9) neurons. (**C**) Average $[Ca^{2+}]_o$ dependent voltage changes sensitive to TTX and $Gd^{3+}$ calculated by subtraction of data in B, and the remaining $[Ca^{2+}]_o$-dependent voltage after application of both blockers (**Figure 6—source data 1**). (**D**) Estimates of the average relative size of the external divalent concentration-dependent NALCN and VGSC currents in neocortical neurons between −100 and −30 mV. NALCN values from **Figure 5G**. The external divalent concentration-dependent VGSC currents were estimated as follows: the products of the VGSC activation and inactivation conductance plots were calculated for $T_{1.1}$ and $T_{0.2}$ using the average Boltzmann curves in **Figure 3**. These were converted to currents (I = driving voltage x conductance), and scaled to match the average TTX-sensitive current at −70 mV. The current generated in $T_{0.2}$ minus that generated in $T_{1.1}$ ($\Delta I_{Ca}$) was plotted against membrane voltage. (**E**) Plot of the average divalent-dependent depolarizing current carried by VGSC derived from D. The change in average resting membrane potential recorded in **Figure 1** is indicated by the gray bar.

The online version of this article includes the following source data for figure 6:

**Source data 1.** Depolarization elicited by switch from $T_{1.1}$ to $T_{0.2}$ that was sensitive to TTX, $Gd^{3+}$, or resistant to both blockers in conventional WT and $Casr^{-/-}$ neurons.

peak nor end current were affected by reduction of the external divalent concentration or by deletion of CaSR (**Figure 4**) over the range of voltages. The currents activated at 60 mV were similarly unaffected (**Figure 4**, Two-way RM ANOVA [(3, 57)=1.347, p=0.2683 and (1, 19)=1.231, p=0.2811, **Table 12**]). These data indicate that VGPCs are not involved in divalent-dependent excitability in neocortical neurons.

**Table 10.** Voltage-gated sodium channel current $V_{0.5}$ for inactivation.

| ANOVA table | SS | DF | MS | F (DFn, DFd) | p Value |
|---|---|---|---|---|---|
| Interaction | 12.00 | 1 | 12.00 | F (1, 18)=0.7743 | p=0.3905 |
| $[Ca^{2+}]_o$ | 3973 | 1 | 3973 | F (1, 18)=256.2 | p<0.0001 |
| Genotype | 27.49 | 1 | 27.49 | F (1, 18)=0.5632 | p=0.4627 |
| Subjects (matching) | 878.5 | 18 | 48.81 | F (18, 18)=3.148 | p=0.0097 |
| Residual | 279.1 | 18 | 15.50 | | |

## VGSCs are the dominant contributor to divalent-dependent currents

To compare the contributions of VGSCs and NALCN to the divalent-dependent depolarization seen in neocortical neurons (*Figure 2*), we measured the size of the currents elicited at −70 mV in neurons following the switch from $T_{1.1}$ to $T_{0.2}$. We used conWT neurons to avoid potential confounding Cre-dependent effects (*Qiu et al., 2011*). Since NALCN is resistant to the VGSC blocker tetrodotoxin (TTX) (*Lu et al., 2007*; *Swayne et al., 2009*) but $Gd^{3+}$ (10 μM) inhibits NALCN and VGSCs (*Elinder and Arhem, 1994*; *Li and Baumgarten, 2001*; *Lu et al., 2009*), we were able to pharmacologically separate the contributions of VGSCs and NALCN to the basal current following the switch from $T_{1.1}$ to $T_{0.2}$ (-31 ± 3 pA, n = 13; *Figure 5A–C*). Addition of a saturating concentration of TTX (1 μM) in $T_{0.2}$ inhibited a persistent inward current within a few seconds in all but one of the recordings (*Figure 5A–C*), consistent with VGSCs contributing to the inward current elicited by $T_{0.2}$. Switching to $T_{1.1}$ plus TTX produced minimal change in the basal current on average (*Figure 5C*). However, in some neurons, $T_{1.1}$ elicited an outward current (*Figure 5A,C*), whereas in others there was an inward current (*Figure 5B,C*) indicating the presence of two types of TTX resistant divalent-sensitive pathways. Presumably, NALCN was contributing to the divalent-dependent TTX-resistant effect observed in *Figure 5A*. Co-application of $Gd^{3+}$ (10 μM) following block of VGSCs with TTX, resulted in a small inward deflection of the average basal current in solution $T_{1.1}$ and largely inhibited sensitivity to concomitant decreases in $[Ca^{2+}]_o$ (*Figure 5A–C*). The reduced sensitivity of neurons to the reduction of $[Ca^{2+}]_o$ in the presence of TTX, suggests that VGSCs are a major contributor to the depolarizing current elicited by low $[Ca^{2+}]_o$. Using serial subtraction of the basal currents (*Figure 5C*), we compared the size of the TTX-sensitive (−28.2 ± 5.3 pA), $Gd^{3+}$-sensitive (−5.7 ± 3.4 pA) and remaining (3.4 ± 2.0 pA) divalent-dependent currents (*Figure 5D*; RM-ANOVA, F (1.495, 17.94)=13.30, p=0.0007, *Table 13*). Multiple comparison testing indicated that the TTX-sensitive divalent-dependent current was greater than the $Gd^{3+}$-sensitive (p=0.039) and remaining divalent-dependent currents (p=0.0009; *Table 14*). Similar differences in the relative sizes of the TTX-, $Gd^{3+}$-, and remainder divalent-dependent basal current currents were also observed in Casr$^{-/-}$ neurons (*Figure 5D*). While there were rare neurons in which there was a larger $Gd^{3+}$-sensitive current (*Figure 5C*) the reduced sensitivity of neurons to the reduction of $[Ca^{2+}]_o$ in the presence of TTX, confirms that VGSCs are the major contributor to the depolarizing current elicited by low $[Ca^{2+}]_o$.

In a fraction of the neurons, an inward deflection of the basal current occurred when external divalent concentration was increased in the presence of TTX (*Figure 5B,C*) which contrasted with the outward current expected from NALCN deactivation (*Figure 5A*). We examined the voltage-dependence of the contributions of VGSCs, NALCN, and this second divalent-dependent TTX-resistant current to better characterize divalent-dependent excitability. We used 50 ms voltage steps between

**Table 11.** Voltage-gated sodium channel current $V_{0.5}$ for activation.

| ANOVA table | SS | DF | MS | F (DFn, DFd) | p Value |
|---|---|---|---|---|---|
| Interaction | 4.814 | 1 | 4.814 | F (1, 17)=0.5668 | p=0.4618 |
| $[Ca^{2+}]_o$ | 834.5 | 1 | 834.5 | F (1, 17)=98.24 | p<0.0001 |
| Genotype | 157.6 | 1 | 157.6 | F (1, 17)=4.813 | p=0.0424 |
| Subjects (matching) | 556.7 | 17 | 32.75 | F (17, 17)=3.855 | p=0.0040 |
| Residual | 144.4 | 17 | 8.494 | | |

**Table 12.** Voltage-gated potassium channel currents at 60 mV.

| ANOVA table | SS | DF | MS | F (DFn, DFd) | p Value |
|---|---|---|---|---|---|
| Interaction | 1.226e-018 | 3 | 4.086e-019 | F (3, 57)=0.2271 | p=0.8772 |
| $[Ca^{2+}]_o$ and time | 7.270e-018 | 3 | 2.423e-018 | F (3, 57)=1.347 | p=0.2683 |
| Genotype | 6.054e-017 | 1 | 6.054e-017 | F (1, 19)=1.231 | p=0.2811 |
| Subjects (matching) | 9.345e-016 | 19 | 4.919e-017 | F (19, 57)=27.33 | p<0.0001 |
| Residual | 1.026e-016 | 57 | 1.800e-018 | | |

−100 and −50 mV and averaged the current over the last 5 ms of the step. Three additional major effects are illustrated by the exemplar current traces (*Figure 5E*). First, in the absence of blockers, the switch from $T_{1.1}$ to $T_{0.2}$ substantially increased the number of large, rapidly inactivating inward currents even at −70 mV following hyperpolarizing steps. Second, in TTX, low $[Ca^{2+}]_o$ increased the linear inward and rectifying outward currents. Third, in the presence of TTX and $Gd^{3+}$ changing between $T_{1.1}$ and $T_{0.2}$ had little effect suggesting $Gd^{3+}$ is blocking both NALCN and the second divalent-dependent TTX-resistant current. These observations were confirmed in the average current-voltage plots (*Figure 5F*) where it is clear that at −80 to −100 mV the major divalent-dependent currents are inward and resistant to TTX and sensitive to $Gd^{3+}$, whereas at −70 to −50 mV the largest divalent-dependent currents are TTX-sensitive. The divalent-dependent effects were calculated by subtracting the currents recorded in $T_{1.1}$ from those in $T_{0.2}$ under control conditions (*Figure 5G*, broken red), in the presence of TTX (broken blue) and TTX plus $Gd^{3+}$ (solid green). The TTX-sensitive (solid red) and $Gd^{3+}$-sensitive (solid blue) divalent-dependent currents were obtained by additional subtraction (broken red minus broken blue and broken blue minus green). The average divalent-dependent current carried by VGSCs only became evident once the neurons were depolarized above −80 mV (*Figure 5G*). The time course of deactivation of the persistent divalent-dependent VGSC currents was observed following hyperpolarization from −70 mV (*Figure 5H*, middle). At more negative potentials, the $Gd^{3+}$-sensitive current accounted for all the divalent-dependent current and traces showed an ohmic voltage dependence (*Figure 5G*). However, the $Gd^{3+}$-sensitive current reversed at −60 mV and outward currents were elicited by steps to −60 and −50 mV that exhibited a voltage-dependent activation and inactivation (*Figure 5H*, right-hand). This is consistent with the $Gd^{3+}$-sensitive current consisting of the sum of NALCN and an outward voltage-dependent current. Assuming conservatively that all of the $Gd^{3+}$-sensitive current at −100 mV could be attributed to NALCN and employing the channel's linear voltage-dependence and zero mV reversal potential (*Lu et al., 2007*; *Lu et al., 2010*), then the amplitude of NALCN currents could be estimated over the voltage range −100 to 0 mV (broken black line, *Figure 5G*). By interpolation (*Figure 5G*, inset), the contribution of NALCN and VGSCs to divalent-dependent currents were equal at −77 mV with the contribution from VGSCs increasing with depolarization. A similar analysis of divalent-dependent currents in *Casr*$^{-/-}$ neurons indicated that the contribution of VGSCs was greater than that of NALCN once membrane potentials were depolarized beyond −72 mV (*Figure 5—figure supplement 1*). These data indicate that divalent-dependent currents around the resting membrane that contribute to divalent-dependent excitability are mainly attributable to VGSCs.

**Table 13.** divalent-dependent basal currents at −70 mV.

| ANOVA table | SS | DF | MS | F (DFn, DFd) | p Value |
|---|---|---|---|---|---|
| Treatment | 6.871e-021 | 2 | 3.435e-021 | F (1.495, 17.94)=13.30 | p=0.0007 |
| Individual (between rows) | 5.669e-022 | 12 | 4.725e-023 | F (12, 24)=0.1828 | p=0.9981 |
| Residual (random) | 6.201e-021 | 24 | 2.584e-022 | | |
| Total | 1.364e-020 | 38 | | | |

**Table 14.** Post hoc testing of divalent-dependent basal currents at −70 mV.

| Sidak's multiple comparisons test | Mean diff. | 95% CI of diff. | Significant? | Summary | Adjusted p value |
|---|---|---|---|---|---|
| TTX sens vs. Gd³⁺ sens | −2.247e-011 | −4.391e-011 to −1.033e-012 | Yes | * | 0.0392 |
| TTX sens vs. Rem | −3.158e-011 | −4.899e-011 to −1.418e-011 | Yes | *** | 0.0009 |
| Gd³⁺ sens vs. Rem | −9.111e-012 | −2.146e-011 to 3.240e-012 | No | ns | 0.1789 |

## Changes in resting potential resulting from lowered divalents are mediated mainly by VGSCs

The complex architecture of neocortical neurons restricted our ability to clamp the membrane potential following the activation of large, rapid VGSC currents. Thus, we re-examined the contribution of VGSCs and NALCN to the depolarizations that mediate divalent-dependent excitability in current clamp recordings from conWT neurons. Consistent with earlier experiments (*Figure 2*), switching from $T_{1.1}$ to $T_{0.2}$ depolarized the membrane potential from −70 mV by 7.2 ± 1.5 mV (n = 12) and increased spontaneous action potential firing in pharmacologically isolated neurons (*Figure 6A,B*). We used TTX and Gd³⁺ to measure the contributions of VGSCs and NALCN respectively to these divalent-dependent depolarizations. TTX blocked action potential generation, as expected, but also hyperpolarized the membrane potential indicating that VGSCs were open in $T_{0.2}$ (*Figure 6A1*) and $T_{1.1}$ (*Figure 6A2*). The switch from $T_{0.2}$ to $T_{1.1}$ in TTX resulted in a hyperpolarization, consistent with divalent-dependent NALCN closure, in some neurons (*Figure 6A* 1 lower trace and B). Other neurons depolarized with the switch to $T_{1.1}$ (*Figure 6A* 2 lower trace and B) consistent with a divalent-dependent outward current similar to that observed in *Figure 5B,C*. On average the divalent-dependent depolarization was almost entirely prevented by TTX or TTX and Gd³⁺ co-application (*Figure 6B*). The amplitude of the divalent-dependent depolarizations in conWT neurons changed with blocker type (1-way RM ANOVA, F (1.219, 13.41)=12.83, p=0.0022, *Table 15*). The TTX-sensitive component was greater than the Gd³⁺-sensitive and the blocker-resistant component (p=0.022 and 0.0028 respectively, *Table 16*). On average VGSCs accounted for 93% of the depolarization that triggers divalent-dependent excitability in WT neurons starting at −70 mV (*Figure 6C*) and we observed a similar pattern in Casr⁻/⁻ neurons (*Figure 6C*).

Next we estimated the average relative contributions of the divalent-dependent NALCN and VGSC currents over a wider voltage range. Ohmic divalent-dependent NALCN currents were extrapolated from −100 mV, where contaminating currents appear minimal (*Figure 5G*) and compared with the divalent-dependent VGSC currents predicted from scaled conductance plots (*Figure 3D*). The VGSC currents were the major contributor to divalent-dependent currents over the −77 to −30 mV voltage range (*Figure 6D,E*). These findings indicate that VGSCs are the predominant contributor to the depolarizations that lead to action potential generation at lower external divalent concentrations (*Figure 6E*, gray bar).

## Discussion

Extracellular calcium concentration regulates both synaptic transmission and intrinsic neuronal excitability, thereby strongly affecting the probability of action potential generation. Consequently, physiological and pathological changes in $[Ca^{2+}]_o$ will impact neuronal computation in a complex manner. We have investigated the mechanisms underlying divalent-dependent changes in intrinsic neuronal excitability and tested if CaSR is transducing decreases in $[Ca^{2+}]_o$ into NALCN-mediated

**Table 15.** divalent-dependent depolarization.

| ANOVA table | SS | DF | MS | F (DFn, DFd) | p Value |
|---|---|---|---|---|---|
| Treatment | 0.0003944 | 2 | 0.0001972 | F (1.219, 13.41)=12.83 | p=0.0022 |
| Individual (between rows) | 0.0001037 | 11 | 9.423e-006 | F (11, 22)=0.6132 | p=0.7982 |
| Residual (random) | 0.0003381 | 22 | 1.537e-005 | | |
| Total | 0.0008361 | 35 | | | |

**Table 16.** Post hoc testing of blocker sensitive fractions of the divalent-dependent depolarization.

| Sidak's multiple comparisons test | Mean diff. | 95% CI of diff. | Significant? | Summary | Adjusted p value |
|---|---|---|---|---|---|
| TTX sens vs. Gd$^{3+}$ sens | 0.006528 | 0.001005 to 0.01205 | Yes | * | 0.0215 |
| TTX sens vs. Rem | 0.007428 | 0.002879 to 0.01198 | Yes | ** | 0.0028 |
| Gd$^{3+}$ sens vs. Rem | 0.0009 | −0.001301 to 0.003101 | No | ns | 0.5311 |

depolarizations to trigger action potentials (*Lu et al., 2010*). We found no evidence that this specific mechanism was active in neocortical neurons (*Figure 2*). Instead, we determined that the vast majority of divalent-dependent neuronal excitability was mediated via VGSCs in three ways. Decreasing the concentration of external divalents activated VGSCs at the resting membrane potential and depolarized the membrane toward the action potential threshold (*Figure 6*). This occurred because the decreased divalent concentration hyperpolarized the VGSC window current toward the membrane potential increasing sodium currents and the likelihood of action potential generation (*Figure 3*). Unexpectedly the deletion of CaSR modulated VGSC gating, decreasing the sensitivity of current activation to depolarization via an unidentified mechanism (*Figure 3*). Deletion of CaSR also indirectly affected action potential generation by modestly hyperpolarizing the membrane potential (*Figure 1*). While the actions of $[Ca^{2+}]_o$ on VGSCs were responsible for the vast majority of the $[Ca^{2+}]_o$-dependent neuronal excitability, using Gd$^{3+}$ we isolated small divalent-dependent inward currents in about half of the neurons (*Figure 5*). These Gd$^{3+}$-sensitive currents presumably reflected activation of NALCN, and were unaffected by CaSR deletion, but their relatively small size compared to TTX-sensitive divalent-dependent inward currents indicate that they would be minor contributors to divalent-dependent excitability compared to VGSCs (*Figures 5* and *6*).

The fractions of the divalent-dependent currents and depolarizations that were sensitive to TTX were surprisingly large compared to those that were Gd$^{3+}$-sensitive (*Figures 5D* and *6C*) indicating the relative importance of VGSC- and NALCN-mediated contributions to divalent-dependent excitability respectively. The resistance of NALCN to TTX (*Lu et al., 2007*; *Swayne et al., 2009*) reassures that the relatively large TTX-sensitive component is due to selective block of VGSC currents. Persistent subthreshold VGSC currents have been shown to determine spiking rates in other central neurons (*Taddese and Bean, 2002*; *Gorelova and Seamans, 2015*) and so the increased VGSC currents we observed in $T_{0.2}$ are well-positioned to explain the increased action potential frequency (*Figure 6*). We are unable to determine from these experiments which neuronal compartment is most affected by the change in $[Ca^{2+}]_o$ (*Gorelova and Seamans, 2015*). However, the physiological impact of VGSC-mediated divalent-dependent excitability may be enormous overall because of the dynamic nature of $[Ca^{2+}]_o$ in vivo where it decreases from basal (1.1–1.2 mM) by 30–80% (*Nicholson et al., 1978*; *Ohta et al., 1997*; *Pietrobon and Moskowitz, 2014*). The overall computational effects of physiological decrements in $[Ca^{2+}]_o$ will be complex because the increased action potential generation due to changes on VGSCs (*Figures 3* and *6*) will be confounded by the impact of reduced $Ca^{2+}$ entry through VACCs (*Hess et al., 1986*; *Weber et al., 2010*; *Williams et al., 2012*), reduced excitatory synaptic transmission (*Neher and Sakaba, 2008*; *Vyleta and Smith, 2011*), and altered CaSR-mediated signaling at the nerve terminal (*Phillips et al., 2008*; *Chen et al., 2010*; *Vyleta and Smith, 2011*).

It remains unclear why NALCN was the dominant effector of divalent-dependent excitability in hippocampal (*Lu et al., 2010*) but not neocortical neurons (*Figure 6*). Could our use of $[Ca^{2+}]_o$ and $[Mg^{2+}]_o$ rather than $[Ca^{2+}]_o$ alone be responsible? We changed divalents simultaneously to provide a strong stimulus to CaSR-signaling and VGSC gating, both of which are sensitive to $[Ca^{2+}]_o$ and $[Mg^{2+}]_o$ (*Frankenhaeuser and Hodgkin, 1957*; *Brown et al., 1993*). Consequently, the same pathways were expected to respond to changes in divalents or $[Ca^{2+}]_o$ alone, since the potentially confounding effects on synaptic transmission were blocked in our experiments. Another difference is that we counted spontaneous action potentials as the main measure of excitability whereas others have focused on action potentials elicited by direct injection. We used spontaneous activity to allow us to isolate the depolarization (*Figure 2*) that was hypothesized to arise from NALCN activation and trigger action potentials following the reduction of external divalent concentration changes (*Figure 2*). Spontaneous and depolarization-elicited action potentials have been recognized as forms of

[Ca$^{2+}$]$_o$-dependent excitability for >60 years (*Frankenhaeuser and Hodgkin, 1957*) and both types of activity were increased here when external divalent concentrations were decreased (*Figure 1* and *Figure 1—figure supplement 2*). Because we observed increased excitability, despite the injection of a current to bring the steady state membrane potential back to that recorded in T$_{1.1}$, mechanisms other than a voltage-independent non-selective cation channel, like NALCN, must have been active (*Figure 1* and *Figure 1—figure supplement 2*). Similarly, the increased spikes elicited by transient current injections in low [Ca$^{2+}$]$_o$ in hippocampal neurons occurred after the steady state membrane potential was set to −80 mV using a longer current injection (*Lu et al., 2010*). The long injection would have reversed the NALCN-mediated depolarization in low [Ca$^{2+}$]$_o$ and so the mechanism by which the increased excitability occurred is unclear. One possible explanation is that at low [Ca$^{2+}$]$_o$ NALCN could have been further activated by shorter depolarizing current injections; however, this is at odds with the lack of voltage-dependence of NALCN (*Lu et al., 2010*). Could NALCN be operating via a different mechanism? One possibility is that NALCN activation is enhancing excitability measured at the soma by enhancing calcium entry into nerve terminals (directly or modifying the action potential waveform and VACC activation) and strengthening excitatory synaptic transmission onto the neuron under study. This would require that the enhancement of synaptic transmission by NALCN be greater than the reduction due to reduced Ca$^{2+}$ entry (*Neher and Sakaba, 2008*) but could be addressed by recording directly from terminals (*Ritzau-Jost et al., 2021*) or by determining if NALCN deletion has the same effect after blocking glutamatergic transmission. However, the loss of NALCN could be contributing to [Ca$^{2+}$]$_o$-dependent changes in excitability independent of a depolarization based on other reports. A number of mechanisms have been postulated to explain how a persistent sodium leak into excitable cells at rest can affect excitability (*Sokolov et al., 2007*). Such mechanisms or other compensatory changes in neuronal function, as observed with null-mutant animals (*Jun et al., 1999*), could arise from the loss of NALCN and possibly contribute to the reduced sensitivity of hippocampal neurons to decreased [Ca$^{2+}$]$_o$ (*Lu et al., 2010*). Lastly, the apparent difference between the studies could reflect different properties of hippocampal and neocortical neurons. While possible it still remains unclear why the deletion of NALCN or UNC-79 completely ablated [Ca$^{2+}$]$_o$-dependent excitability in hippocampal neurons (*Lu et al., 2010*) since these neurons contain VGSCs that retain sensitivity to changes in [Ca$^{2+}$]$_o$ (*Isaev et al., 2012*). However, if the UNC79-UNC80-NALCN pathway modulates VGSC function this could explain how loss of NALCN or UNC-79 could delete acute divalent-dependent changes in VGSC function and excitability. NALCN appears to transduce [Ca$^{2+}$]$_o$- and G-protein-dependent excitability in other neurons (*Philippart and Khaliq, 2018*) but GPCRs other than CaSR may be involved (*Kubo et al., 1998*; *Tabata and Kano, 2004*) and under certain conditions Ca$^{2+}$ directly blocks NALCN (*Chua et al., 2020*). Further characterization of the UNC79-UNC80-NALCN signaling pathway is essential given the major changes in neurological function that have been described following mutations of NALCN or upstream co-molecules such as UNC79 and UNC80 (*Stray-Pedersen et al., 2016*; *Bourque et al., 2018*; *Kuptanon et al., 2019*).

In a small fraction of the neocortical neurons (*Figures 5* and *6*) there was a modest inward current or depolarization with the lowering of extracellular divalent concentration once VGSCs had been blocked. In a few cases, they were sensitive to 10 μM Gd$^{3+}$ consistent with a NALCN-mediated effect and those that were resistant were consistent with other divalent-dependent non-selective cation channels (*Ma et al., 2012b*). However, deletion of CaSR did not decrease divalent-dependent depolarizations and after membrane potential matching did not impact divalent-dependent excitability (*Figures 2E* and *6B*). While CaSR-NALCN signaling did not contribute to divalent-dependent excitability in neocortical neurons (*Figures 2* and *6*) it was clear that Casr$^{-/-}$ neurons were substantially less sensitive to changes [Ca$^{2+}$]$_o$ (*Figure 1A,B*). The reduced [Ca$^{2+}$]$_o$ sensitivity in these neurons is attributable to the combination of altered VGSC gating (*Figure 3E,F*) and the hyperpolarized RMP (*Figure 1E*). Although CaSR did not affect the amplitude of the shift in V$_{0.5}$ following the switch to T$_{0.2}$, the gating characteristics of VGSC activation was depolarized by loss of CaSR (*Figure 3EF*). Could CaSR stimulation activate G-proteins and regulate the V$_{0.5}$ for VGSC currents (*Figure 3E,F*)? In neocortical neurons, G-protein activation hyperpolarized VGSC gating and this was blocked by GDPβS (*Mattheisen et al., 2018*) which is inconsistent with the effect we observed here. Other possible explanations are that CaSR could regulate VGSC subunit expression or post translational modification (*Cantrell et al., 1996*; *Zhang et al., 2019*), and this may represent a compensatory mechanism similar to that observed with other mutant mouse models (*Jun et al., 1999*). Loss of

CaSR also hyperpolarized the neocortical neurons (*Figure 1I*) and this may have been due to decreased function of depolarizing components or stimulation of hyperpolarizing elements. There are a number of candidate channels and pumps that have been shown to regulate the RMP in cortical neurons (*Tavalin et al., 1997*; *Talley et al., 2001*; *Bean, 2007*; *Harnett et al., 2015*; *Hu and Bean, 2018*). The changes in VGSC gating and RMP in *Casr⁻/⁻* neurons may be attributable to homeostatic mechanisms that compensate for perturbations in network activity and have been observed in central and peripheral neurons (*Turrigiano, 2008*).

Overall, our studies indicate that divalent-dependent excitability in neurons is largely attributable to actions of extracellular calcium on the VGSC function. Given the dynamic nature of brain extracellular calcium, this mechanism is likely to impact neuronal signaling greatly under physiological and pathological conditions. CaSR-dependent reduction of VGSC sensitivity to membrane potential adds further complexity to extracellular calcium signaling and identifies another potential mechanism by which CaSR stimulation may influence neuronal death following stroke and traumatic brain injury (*Kim et al., 2013*; *Hannan et al., 2018*).

# Materials and methods

## Key resources table

| Reagent type (species) or resource | Designation | Source or reference | Identifiers | Additional information |
|---|---|---|---|---|
| Gene (*M. musculus*) | Casr | GenBank | Casr | |
| Strain, strain background (*M. musculus*) | Mouse wild-type strain C57BL/6J × 129×1 | The Jackson Laboratory | RRID:MGI:5652742 | |
| Genetic reagent, strain background (*M. musculus*) | Mouse expressing Nestin-cre mutation | The Jackson Laboratory as used in *Sun et al., 2018* | Stock No. 003771 | C57/BL6J and 129S4 background strain |
| Genetic reagent, strain background (*M. musculus*) | Mouse with Lox mutation to delete exon 7 of Casr | Laboratory of Dr. Wenhan Chang, UCSF (*Chang et al., 2008*) | *Casr*fl/fl | C57/BL6J and 129S4 background strain |
| Sequence-based reagent | Casr | Applied Biosystems | Mm00443377_m1 | Quantitative PCR Mouse probe set |
| Sequence-based reagent | Actb | Applied Biosystems | Mm04394036_g1 | Quantitative PCR Mouse probe set |
| Sequence-based reagent | Nes-Cre1 primer | IDT | GCAAAACAGGCTCTAGCGTTCG | |
| Sequence-based reagent | Nes-Cre2 primer | IDT | CTGTTTCACTATCCAGGTTACGG | |
| Sequence-based reagent | P3U primer | IDT | TGTGACGGAAAACATACTGC | |
| Sequence-based reagent | Lox R primer | IDT | GCGTTTTTAGAGGGAAGCAG | |

## Genotyping and CaSR mutant mice

ConWT animals were obtained from an established colony consisting of a stable strain of C57BL/6J and 129 × 1 mice. The *Casr⁻/⁻* mice were generated by breeding floxed Casr (*Chang et al., 2008*) and nestin Cre mice (B6.Cg-Tg (Nes-cre)1Kln/J, The Jackson Laboratory) as described previously (*Sun et al., 2018*). The lox sites were positioned to delete Casr exon seven which resulted in the loss of Casr expression (*Chang et al., 2008*) and the nestin promoter was designed to ensure floxing occurred in neuronal and glial precursors. The *NesCre* mice were generated by crossing mice that did not contain the flox Casr mutation but did express the nestin Cre mutation. The *Casr⁻/⁻* and *Nes-Cre* mice were all generated using a background C57BL/6J and 129S4 strain. Tail DNA extraction was performed using the Hot Shot Technique with a 1–2 hr boil (*Montero-Pau et al., 2008*). The

presence or absence of the flox Casr mutation and Cre mutation were confirmed by PCR for each mouse. MoPrimers used for cre PCR were: Nes-cre1: GCAAAACAGGCTCTAGCGTTCG, Nes-cre2: CTGTTTCACTATCCAGGTTACGG; run on a 1% agarose gel. Primers for lox PCR were: P3U: TGTGACGGAAAACATACTGC, Lox R: GCGTTTTTAGAGGGAAGCAG; run on a 1.5% agarose gel (*Chang et al., 2008*). Successful deletion of Casr in the neocortical cultures was confirmed by measuring mRNA expression levels with the QuantStudio12K Flex Real-time PCR System (Applied Biosystems) and the TaqMan mouse probe set to Casr (Mm00443377_m1) with ActB (Mm04394036_g1) as the endogenous control (*Figure 1—figure supplement 1*). The paper describes experiments comparing the effects of CaSR deletion using the $Casr^{-/-}$ mice. After confirming that conWT and $Nes^{-Cre}$ neurons responded similarly to changing external divalents (*Figure 1A–D*) we used $Nes^{Cre}$ neurons and $Casr^{-/-}$ neurons to examine if Casr was responsible for the sensitivity to extracellular divalents. This comparison avoided possible confounding cre-dependent effects (*Qiu et al., 2011*). In later experiments, we used conventional WT to ensure that our measurements of the relative size of the effect of VGSC and NALCN were not impacted by cre-dependent effects (*Qiu et al., 2011*).

## Neuronal cell culture

Neocortical neurons were isolated from postnatal day 1–2 mouse pups of either sex as described previously (*Phillips et al., 2008*). All animal procedures were approved by V.A. Portland Health Care System Institutional Animal Care and Use Committee in accordance with the U.S. Public Health Service Policy on Humane Care and Use of Laboratory Animals and the National Institutes of Health Guide for the Care and Use of Laboratory Animals. The active protocols covering this work are 4254–19 and 4359–20. Animals were decapitated following induction of general anesthesia with isoflurane and then the cerebral cortices were removed. Cortices were incubated in trypsin and DNase and then dissociated with a heat polished pipette. Dissociated cells were cultured in MEM plus 5% FBS on glass coverslips. Cytosine arabinoside (4 µM) was added 48 hr after plating to limit glial division. Cells were used, unless otherwise stated after ≥14 days in culture.

## Electrophysiological recordings

Cells were visualized with a Zeiss IM 35 inverted microscope. Whole-cell voltage-and current-clamp recordings were made from cultured neocortical neurons using a HEKA EPC10 amplifier. Except where stated in the text, extracellular Tyrodes solution contained (mM) 150 NaCl, 4 KCl, 10 HEPES, 10 glucose, 1.1 $MgCl_2$, 1.1 $CaCl_2$, pH 7.35 with NaOH. Calcium and magnesium concentrations were modified as described in the Figure legends. The CaSR and surface charge screening are both sensitive to $Ca^{2+}$ and $Mg^{2+}$ with $Ca^{2+}$ being two to three times more effective in both processes (*Frankenhaeuser and Hodgkin, 1957*; *Brown et al., 1993*). We modified the divalent concentrations simultaneously to utilize a greater fraction of the dynamic range of the phenomenon under study and to avoid irreversible changes that can occur in $Ca^{2+}$-free solutions (*Frankenhaeuser and Hodgkin, 1957*). Synaptic transmission was blocked by the addition of (in µM) 10 CNQX, 10 Gabazine, and 50 APV to the bath solution. Most recordings were made using a potassium gluconate intracellular solution containing (mM) 135 K-gluconate, 10 HEPES, 4 $MgCl_2$, 4 NaATP, 0.3 NaGTP, 10 phosphocreatine disodium, pH 7.2 with KOH hydroxide. In nucleated patch experiments, the pipette solution contained (in mM) 113 Cesium methane sulfonate, 1.8 EGTA, 10 HEPES, 4 $MgCl_2$, 0.2 $CaCl_2$, 4 NaATP, 0.3 NaGTP, 14 phosphocreatine disodium, pH 7.2 with TEA hydroxide. Electrodes used for recording had resistances ranging from 2 to 7 MΩ. Voltages have been corrected for calculated liquid junction potentials (JPCalc, Professor P. H. Barry) and were 14 or 15 mV for all recordings. All experiments were performed at room temperature (21–23°C).

## Data acquisition and analysis

Whole-cell voltage-and current-clamp recordings were made using a HEKA EPC10 USB amplifier, filtered at 2.9 kHz using a Bessel filter, and sampled at 20 kHz during injection protocols and 10 kHz during continuous acquisition. Analysis was performed using Igor Pro (Wavemetrics, Lake Oswego, OR) and Minianalysis (Synaptosoft). Data values are reported as mean ± SEM. Statistical tests were performed using GraphPad Prism (6) and p-values<0.05, 0.01, 0.001, and 0.0001 were indicate with *, **, ***, and ****. All post-hoc tests were Sidak compensated for multiple comparisons. Data were log-transformed to improve normalization in *Figure 2D*. To ensure non-zero values, minimize bias,

and allow logarithmic transformation, each action potential frequency measurement was increased by 0.02 as the duration of the $T_{1.1}$ recording at $-70$ mV was 50 s.

## Solution application

Solutions were applied by gravity from a glass capillary (1.2 mm outer diameter) placed ~1 mm from the neuron under study. Solutions were switched manually using a low dead volume manifold upstream of the glass capillary. CNQX and Gabazine were supplied by Abcam. KB-R7943 Mesylate was supplied by Tocris. Creatine Phosphate was supplied by Santa Cruz Biotech. Cinacalcet was supplied by Toronto Research Chemicals and Tetrodotoxin by Alomone Other reagents were obtained from Sigma-Aldrich.

## Acknowledgements

This work was supported by grants awarded by U.S. Department of Veterans Affairs (BX002547) and NIGMS (GM134110) to SMS. We thank Dr Wenyan Chen for performing the experiments on potassium channel currents and Dr Glynis Mattheisen, Dr Brian Jones, Ms Natasha Baas-Thomas, and Ms Maya Feldthouse for helpful discussion and comments on the manuscript. Thanks to Drs Chris Harrington and Brittany Daughtry of the OHSU Gene Profiling Shared Resource who performed RNA isolation, quality assessments, and qPCR assays. The authors declare no competing financial interests. The contents do not represent the views of the U.S. Department of Veterans Affairs or the United States Government.

## Additional information

### Funding

| Funder | Grant reference number | Author |
|---|---|---|
| U.S. Department of Veterans Affairs | BX002547 | Stephen M Smith |
| National Institute of General Medical Sciences | GM134110 | Stephen M Smith |
| U.S. Department of Veterans Affairs | IK6BX004835 | Wenhan Chang |
| U.S. Department of Veterans Affairs | BX003453 | Wenhan Chang |

The funders had no role in study design, data collection and interpretation, or the decision to submit the work for publication.

### Author contributions

Briana J Martiszus, Formal analysis, Investigation, Writing - original draft; Timur Tsintsadze, Formal analysis, Investigation; Wenhan Chang, Resources, Methodology; Stephen M Smith, Conceptualization, Data curation, Formal analysis, Supervision, Funding acquisition, Methodology, Project administration, Writing - review and editing

### Author ORCIDs

Stephen M Smith ![ORCID] https://orcid.org/0000-0002-0331-7615

### Ethics

Animal experimentation: All animal procedures were approved by V.A. Portland Health Care System Institutional Animal Care and Use Committee in accordance with the U.S. Public Health Service Policy on Humane Care and Use of Laboratory Animals and the National Institutes of Health Guide for the Care and Use of Laboratory Animals. The active protocols covering this work are 4254-19 and 4359-20.

Decision letter and Author response
Decision letter https://doi.org/10.7554/eLife.67914.sa1
Author response https://doi.org/10.7554/eLife.67914.sa2

## Additional files

### Supplementary files
• Transparent reporting form

### Data availability

All data generated are in the manuscript and supporting files. Source provided for Figures 1, 2, and 6 in the manuscript.

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
