## [Decision Letter]

**Acceptance summary:**

This manuscript presents some interesting and important results about how reduction of external divalent ions enhances the intrinsic excitability of cortical neurons. The authors test the idea that the effect of reducing external calcium to increase neuronal excitability is due to enhancement of sodium leak channel (NALCN) current via a pathway initiated by calcium-sensing receptor (CaSR), a G-protein-coupled receptor. The authors conclude that in cultured cortical neurons, the major effects of reducing divalent ions on excitability are not mediated by CaSR-mediated changes in NALCN current but rather by altered voltage-dependence of voltage-dependent sodium current. While contrary to expectations, these key conclusions are nonetheless in agreement with the classical view. A better understanding of how neuronal excitability is affected by changes in extracellular calcium is crucial given that large changes in extracellular calcium are thought to drive circuit and behavioral changes under pathological conditions.

**Decision letter after peer review:**

[Editors’ note: the authors submitted for reconsideration following the decision after peer review. What follows is the decision letter after the first round of review.]

Thank you for submitting your work entitled "CaSR modulates sodium channel-mediated Ca^2+^-dependent excitability" for consideration by *eLife*. Your article has been reviewed by 3 peer reviewers, one of whom is a member of our Board of Reviewing Editors, and the evaluation has been overseen by Kenton Swartz as Senior Editor. The following individuals involved in review of your submission have agreed to reveal their identity: Stephen Williams (Reviewer #3).

Our decision has been reached after consultation between the reviewers.

The reviewers have agreed that the study addresses an important question and that the findings are potentially interesting. Nevertheless, several major issues remain, in particular the potential developmental effects of the lack of expression of CaSR, voltage-clamp analysis being performed in very young neurons, as well as a lack of confirmation of the knock-out of CaSR protein.

Based on the discussions and the individual reviews below, we regret to inform you that your work will not be considered further for publication in *eLife*, at least for the present submission. However, provided that the major concerns of the reviewers could be fully addressed involving additional experiments, we encourage resubmission of the revised manuscript.

*Reviewer #1:*

This study re-examines the contribution of sodium leak channel (NALCN) downstream to the activation of calcium sensing receptor (CaSR) and the UNC79-UNC80 complex in regulating extracellular calcium-dependence of membrane excitability. Recording from cultured neocortical neurons, contrary to the previously reported role for CaSR in hippocampal neurons, the authors find that robust extracellular calcium-dependent changes in excitability remains in CaSR knockout neurons specifically after correcting for the hyperpolarized membrane potential in CaSR KO neurons. Moreover, a switch to bath solution containing reduced divalents depolarizes both WT and Casr-/- neurons, where VGSC gating is shifted. Interestingly, the shift in VGSC gating observed in Casr-/- neurons is not accompanied by changes in NALCN sensitivity to extracellular calcium. It is concluded that extracellular calcium-dependence of membrane excitability is largely attributed to changes in VGSC gating rather than being mediated via NALCN through CaSR. While the topic is important, the study touches on a controversial mechanism. Therefore, to deliver a compelling conclusion, there are a number of issues that require careful consideration.

1. One should directly confirm by immunostaining or western blots that the expression of CaSR protein is actually lost in Casr-/- neurons used for recordings.

2. Figure 3. While it is appreciated that the voltage clamp is difficult in older neurons, it is highly questionable that the expression of channels and regulatory proteins of relevance are comparable between DIV 2-3 neurons vs. DIV 14-28 neurons. Although not definitive, at the least, the authors should provide experimental support for similar levels of protein expression of VGSC, CaSR, NALCN and VGPC in cortical neurons at two different culture time points.

3. Figure 3A. Representative traces for creCasr-/- cells should be shown also.

4. Figure 5. By what basis control WT neurons used here are not those expressing Cre that have been used in the previous figures? Figure 1A-C suggests that the expression of Cre may cause subtle non-specific effects. Thus the interchangeable use of conWT and creWT is a source of concern.

5. Figure 5A-D. It would be informative to reverse the order of addition of VGSC and NALCN blockers – Gd^3+^ first, followed by TTX, to confirm that the minimal contribution of NALCN to extracellular Ca-dependent depolarization is not dependent on prior blockade of VGSC.

6. Figure 5F, G. It is difficult to discriminate the data between different groups as shown.

7. Figure 6. Which WT neurons were used as controls here?

*Reviewer #2:*

This paper describes that changing the extracellular Ca modulates activation of Ca-sensing receptors (CaSRs) and changes the voltage-dependence of voltage gated Na channels. This is in contrast to the previous postulate that Ca-sensing receptors modulate leak currents. The results are interesting and important, but there are several issues in this study. Also, the papers are a bit difficult to read, because of the way of presentation.

1. Although there is an effect of the CaSR KO neurons, the effect of extracellular Ca itself on Na channel activation seems more drastic. Is it possible that the surface charge screening effects are more dominant?

2. Although voltage clamp of Na channels is difficult, some of the traces (Figure 5) seem to indicate that voltage clamp is not sufficient. Is it possible to use TTX to reduce the current amplitudes and improve the clamp for Figure 3?

3. I do not see the difference between WT and CaSR KO in Figure 1A, which seems to suggest that CaSRs are not really important for the excitability. Perhaps, show the individual data in Figure 1D?

4. Intracellular mechanisms connecting between CaSRs and Na channels are unclear.

5. Overall there are many superimposed traces in the figures, which makes me difficult to understand (Figure 5, for example). Also, it is helpful if the authors draw some scheme at the end of the paper for readability.

*Reviewer #3:*

This well written manuscripts describes electrophysiological recordings from cultured neocortical neurons to examine in control and knockout animals (Casr -/-) the mechanism(s) underlying the augmentation of action potential firing rate by altering the extracellular cation concentration. The topic is of general interest. The manuscript supplies novel and interesting new data. Methodologically the approach taken by the authors is generally sound. The conclusions that the authors make from their data also is logical and justified. I was particularly taken by the observation that the sodium window current plays a significant role in the enhancement of action potential firing. Judging the first two figures of the paper, there seems to be striking differences in the waveforms of action potentials recorded from Casr positive and negative neurons. Do the authors have an explanation for this, does for example Casr influence the developmental trajectory of neurons in culture, in terms of maturation of the expression of voltage gated ion channels? I think the manuscript would benefit from analysis that indicates at given stages of culture that current densities are equivalent etc, which would allow clearer comments to be made by the authors. My main problem with the work lies in the voltage-clamp analysis. The authors indicate in mature cultures that inward currents cannot be voltage-controlled. They therefore turn to analysis of neurons that have been cultured for only a few days. I am not sure that this is the right thing to do – as alluded to above. I would suggest that the authors consider employing recording approaches, such as nucleated patch, that would allow near perfect voltage control. This would allow analysis at the same developmental stage, and also if made from various ages in culture the charting of the developmental profile.

[Editors’ note: further revisions were suggested prior to acceptance, as described below.]

Thank you for submitting your article "CaSR modulates sodium channel-mediated Ca^2+^ -dependent excitability" for consideration by *eLife*. Your article has been reviewed by 3 peer reviewers, one of whom is a member of our Board of Reviewing Editors, and the evaluation has been overseen by Kenton Swartz as the Senior Editor. The reviewers have opted to remain anonymous.

Essential Revisions:

As indicated in the public comments, the authors need to systematically reword the presentation, starting with the title, to reflect the fact that all their experiments examined effects of reducing both Ca and Mg, and it is therefore incorrect to state the changes reflect "[Ca^2+^]-dependent" effects as is done throughout. There were also many places where the wording (and sometimes the order of presentation) seemed unnecessarily convoluted, which made it difficult to understand exactly what the authors meant by a particular sentence. The data in the paper are very interesting and the ability of the reader to digest the results and their meaning would be improved if the presentation were clearer in some places.

Following points in particular require careful consideration:

Title: "CaSR modulates sodium channel-mediated Ca^2+^-dependent excitability". The title is misleading in two ways. First, the main point of the paper is that the most important effects of reducing divalents are NOT mediated by CaSR receptors. Whatever effects on excitability are seen in the CaSR-/- neurons probably reflect homeostatic changes rather than direct function of CaSR receptors, so even the statement that CaSR might "modulate" effects is questionable. In any case, such modulation is a very much secondary point of the paper, not the main one. Second, as already pointed out, the authors looked at changes in response to changing both Ca and Mg and can't say any of the key phenomena examined in the paper are "Ca^2+^-dependent". Something like "Enhanced excitability of cultured cortical neurons in low-divalent solutions is mediated mainly by altered voltage-dependence of voltage-dependent sodium channels" would better capture the content and conclusions of the paper.

Abstract: Abstract was not understandable until after reading the paper. The authors summarize the key results as follows " Here we report that action potential (AP) firing rates increased in wild-type (WT), but not CaSR null mutant (Casr-/-) neocortical neurons, following the switch from physiological to reduced Ca ^2+^-containing Tyrode. However, after membrane potential correction, action potential firing increased similarly in both genotypes inconsistent with CaSR regulation of NALCN." It was completely unclear whether the authors were saying that reduction of calcium did or did not enhance excitability in the CaSR-/- cells, the central question of the paper. The first sentence says "no" and the second sentence says "yes", but only after a "membrane potential correction" that is completely undefined. It is also completely unclear until after reading the paper that "action potential firing" refers to spontaneous firing at the resting potential. As it turns out, a clearer description of the results would be something along the lines of: "CaSR-/- neurons had more negative resting potentials than control neurons and unlike control neurons did not fire spontaneously either in normal or reduced calcium solutions. However action potential firing in response to depolarizing current injections was enhanced in CaSR-/- neurons by reduction of divalent cations to a similar degree as in control neurons."

p. 3 "Classical studies proposed that the mechanism underlying [Ca^2+^]o-dependent excitability centers on voltage-gated sodium channel (VGSC) sensitivity to extracellular Ca^2+^. External Ca^2+^ was proposed to interact with local negative charges on the extracellular face of the membrane or ion channels thereby increasing the potential field experienced by VGSCs and reducing the likelihood of VGSC activation at the resting membrane potential (Frankenhaeuser and Hodgkin, 1957; Hille, 1968)". Frankenhaeuser and Hodgkin only looked at shifts of Na channel gating in voltage clamp experiments and did not report firing of the cells at the resting potential, as the reader might conclude from this. In fact in their discussion they conclude because of the shift in inactivation, axons in low Ca would be inexcitable. A more accurate summary might be something like "Reduced external Ca^2+^ was proposed to shift the effective voltage-dependent gating of the sodium conductance in the hyperpolarizing direction by reducing the screening of local negative charges on the extracellular face of the membrane by external Ca^2+^ (Frankenhaeuser and Hodgkin, 1957; Hille, 1968)".

p. 11: "[Ca^2+^]o reduction and CaSR hyperpolarized this region of overlap towards the RMP (Figure 3 F,G insets) increasing the likelihood that persistent VGSC currents were activated at resting membrane potential and therefore contributing to [Ca^2+^]o-dependent excitability."

The rationale for saying that [Ca^2+^]o reduction (or really divalent reduction) hyperpolarized the region of overlap is clear. However, the rationale connecting the experimental results with saying that "CaSR hyperpolarized the region of overlap" is very vague.

p. 14 paragraph heading "VGSCs are the dominant contributor to [Ca^2+^]o-dependent excitability". This statement seems almost meaningless. There is no excitability of any kind without VGSCs. The meaning of "[Ca^2+^]o-dependent excitability" here (and it is used many other times) is not clear. Perhaps the authors mean "changes in excitability produced by changes in divalent concentrations". In any case, the paragraphs to follow are focused mainly on how changes in resting potential are mediated, not really on excitability. So the meaning is something like "Changes in resting potential resulting from lowered divalents are mediated mainly by VGSCs".

p. 16: "We have investigated the mechanisms underlying [Ca^2+^]o-dependent changes in intrinsic neuronal excitability and tested if CaSR is transducing decreases in [Ca^2+^]o into NALCN-mediated depolarizations to trigger action potentials (Lu et al., 2010; Philippart and Khaliq, 2018). We found no evidence that this specific mechanism was active in neocortical neurons (Figure 2)."

Nothing in the Philippart and Khaliq paper examined whether or not CaSRs were involved in control of NALCN by divalents. Their paper concerned a completely different regulation of NALCN current by GABAB and dopamine receptors.

p. 16 "CaSR also indirectly affected action potential generation by modestly depolarizing the membrane potential (Figure 1)." The authors have no evidence that stimulating CaSRs depolarizes the resting potential. What they mean is that CaSR-/- neurons have hyperpolarized resting potentials compared to WT neurons. They have no evidence of whether this reflects a loss of a normal depolarizing effect of CaSR stimulation or a homeostatic effect of development of the neurons without CaSRs.

p. 21 "Voltages have been corrected for liquid junction potentials." The value used for the corrections should be given, as the exact membrane potentials are quite important for the effects that are shown (especially the firing at the resting potential).

The switch from the use of CreWT to ControlWT lacking the expression of Cre recombinase still remains to be justified. The authors reason that the switch has been made to ControlWT given that they have shown that calcium-dependence of excitability is not mediated by CaSR-modulation of NALCN but by VGSC, and also in order to avoid any confounds presented by the exogenous expression of Cre. Nevertheless, the logic is not clear. The conclusion of the apparent lack of contribution of CaSR-modulation of NALCN has been reached by the use of Cre expressing neurons. The authors should explicitly state whether or not there were any differences in the electrophysiological properties of conWT and creWT neurons and if so, explain what they are, perhaps in the methods.

There were many typos in the paper.

Throughout the text, there seemed to be a global replacement of "T1.1" by a space, which made a lot of sentences hard to read. For example, p. 12 " Switching to plus TTX produced minimal change in the basal current on average (Figure 5C). However, in some neurons elicited an outward current (Figure 5A,C) whereas in others there was an inward current (Figure 5B,C)…" It is assumed that there are two missing "T1.1"'s.

p.11 D,E). The reduction in [Ca^2+^]o (F (1, 18) = 56, P < 0.0001) left- shifted V0.5 (2-way RM ANOVA,

---

## [Author Response]

[Editors’ note: the authors resubmitted a revised version of the paper for consideration. What follows is the authors’ response to the first round of review.]

Reviewer #1:1. One should directly confirm by immunostaining or western blots that the expression of CaSR protein is actually lost in Casr-/- neurons used for recordings.

We have used quantitative RT PCR to determine expression levels of the Casr in wild-type (conventional and cre expressing) and Casr-/- cultured neocortical neurons. We examined expression levels in 6 different cultures (in triplicate) for each of the 3 genotypes. We used actin expression as a positive control. Our data (Figure 1 Supplementary figure) show ≥98% reduction in Casr expression which confirms other studies that used thisknockout mouse and nestin cre-recombinase to delete Casr in neurons. reduction in Casr expression which confirms other studies that used this knockout mouse and nestin cre-recombinase to delete Casr in neurons.

2. Figure 3. While it is appreciated that the voltage clamp is difficult in older neurons, it is highly questionable that the expression of channels and regulatory proteins of relevance are comparable between DIV 2-3 neurons vs. DIV 14-28 neurons. Although not definitive, at the least, the authors should provide experimental support for similar levels of protein expression of VGSC, CaSR, NALCN and VGPC in cortical neurons at two different culture time points.

Two referees pointed out that developmental changes in VGSC could have confounded previous voltage clamp experiments. We addressed this problem using nucleated patches (Figure 3) from neurons of the same age used in other parts of the manuscript. We also confirmed that the voltage dependence (V_0.5_) of VGSC activation and inactivation was stable over the time range of our experiments.

3. Figure 3A. Representative traces for creCasr-/- cells should be shown also.

This figure now replaced with nucleated patch experiments and representative traces added for *Casr^-/-^* cells.

We have added these representative traces.

4. Figure 5. By what basis control WT neurons used here are not those expressing Cre that have been used in the previous figures? Figure 1A-C suggests that the expression of Cre may cause subtle non-specific effects. Thus the interchangeable use of conWT and creWT is a source of concern.

Our main hypothesis was that CaSR regulates NALCN and thereby impact calcium dependent excitability. Studies shown in Figures 1-3 indicated that calcium dependence of excitability was not mediated by CaSR-modulation of NALCN, but rather via VGSCs. Once we had addressed the impact of CaSR (after Figure 4) we switched back to using a regular WT to avoid any possible confounding due to the presence of cre. We have added mention of this to the manuscript (P. 12) to underline this point.

CaSR did shift VGSC gating in Figure 3 and this partially accounted for the different excitability of creWT and cre-Casr-/- neurons in experiments examining what happens at the resting membrane potential. Since the RMP in *Casr^-/-^* neurons was more negative this also contributed to the observed difference in excitability between genotypes. This effect was absent once the membrane potential was slightly offset to -70 mV to reduce the likelihood of RMP confounding the result. The experiments described in Figures 5 and 6 were performed to examine the relative contribution of VGSC and NALCN. We expected to see a contribution by NALCN that was independent of CaSR and chose the conventional WT as we hoped to avoid the small potential confounder that the cre mutation was interfering with calcium-dependent excitability. The *Casr^-/^^-^
*experiments were performed because we were surprised (because of the work by the Ren Lab) to see no obvious sign of NALCN contributing in the WT neurons and hoped to find some explanation here. I have modified Figure 5 in attempt to make the rationale for the experimental design more clear. The *Casr^-/-^* data has been extracted and placed in a supplementary figure.

5. Figure 5A-D. It would be informative to reverse the order of addition of VGSC and NALCN blockers – Gd^3+^ first, followed by TTX, to confirm that the minimal contribution of NALCN to extracellular Ca-dependent depolarization is not dependent on prior blockade of VGSC.

The careful work by others demonstrating NALCN does not possess the TTX binding site and the demonstrations that TTX is ineffective against NALCN in expression systems (Lu et al., 2007; Swayne et al., 2009) reassures that the small action of Gd^3+^ in the presence of TTX is unlikely to be an artefact due to a non-selective action of TTX. Unfortunately, Gd^3+^ blocks VGSCs and so the application of Gd^3+^ alone is expected to block both VGSCs and NALCN. We have edited the manuscript to make this clear (P. 12, Since NALCN is resistant to the VGSC blocker tetrodotoxin (TTX) (Lu et al., 2007; Swayne et al., 2009) but Gd^3+^ (10 µM) inhibits NALCN and VGSCs (Elinder and Arhem, 1994; Li and Baumgarten, 2001; Lu et al., 2009)). However, we also performed the requested experiments to check if Gd^3+^ was selective under our experimental conditions. Using a similar experimental design to that in Figure 5, Gd^3+^ application inhibited the baseline [Ca^2+^ ]_o_-dependent current by a substantial amount if it was applied in the absence of TTX . Gd^3+^ also reversibly reduced the probability of VGSC activation. Given the valence of Gd^3+^ and the sensitivity of VGSC to charged particles this is unsurprising.

6. Figure 5F, G. It is difficult to discriminate the data between different groups as shown.

Apologies. The figure has been redrawn to improve clarity.

7. Figure 6. Which WT neurons were used as controls here?

For the reasons given above (question 4) we used the conventional WT.

Reviewer #2:This paper describes that changing the extracellular Ca modulates activation of Ca-sensing receptors (CaSRs) and changes the voltage-dependence of voltage gated Na channels. This is in contrast to the previous postulate that Ca-sensing receptors modulate leak currents. The results are interesting and important, but there are several issues in this study. Also, the papers are a bit difficult to read, because of the way of presentation.1. Although there is an effect of the CaSR KO neurons, the effect of extracellular Ca itself on Na channel activation seems more drastic. Is it possible that the surface charge screening effects are more dominant?

We agree that the effect on VGSC is the dominant one and modified the manuscript to ensure this is clearer.

2. Although voltage clamp of Na channels is difficult, some of the traces (Figure 5) seem to indicate that voltage clamp is not sufficient. Is it possible to use TTX to reduce the current amplitudes and improve the clamp for Figure 3?

The currents were already very small but we have used nucleated patches to address reviewers’ concerns about the VGSC currents (Figure 3).

3. I do not see the difference between WT and CaSR KO in Figure 1A, which seems to suggest that CaSRs are not really important for the excitability. Perhaps, show the individual data in Figure 1D?

In Figure 1A the number of action potentials in T0.2 is reduced by severalfold in the CaSR KO (red) middle row compared to the two wild types (black and blue). This is emphasized in 1B where the average responses of individual neurons is shown as well as the grand means. This is consistent with the initial hypothesis. However, once adjustments were made for RMP the differences dissipated (Figure 2). In Figure 3 we show a difference in the properties of VGSC currents that indicates one way in which CaSR deletion impacts the calcium sensitivity of VGSCs. The RMP is the other change dependent on CaSR that impacts calcium dependent excitability.

4. Intracellular mechanisms connecting between CaSRs and Na channels are unclear.

We have not yet addressed how CaSR affects VGSC properties experimentally but hope to address this going forwards. I have rewritten the MS to make this clear.

5. Overall there are many superimposed traces in the figures, which makes me difficult to understand (Figure 5, for example). Also, it is helpful if the authors draw some scheme at the end of the paper for readability.

Apologies for overlapping traces. We have expanded the panels to improve readability. I have changed the last panel of Figure 6 to demonstrate the relative size of the contributions of VGSC and NALCN to calcium dependent excitability. In addition we have expanded the discussion on how NALCN might influence excitability in other settings if it does not seem to be regulated by CaSR as hypothesized by Ren and colleagues.

Reviewer #3:This well written manuscripts describes electrophysiological recordings from cultured neocortical neurons to examine in control and knockout animals (Casr -/-) the mechanism(s) underlying the augmentation of action potential firing rate by altering the extracellular cation concentration. The topic is of general interest. The manuscript supplies novel and interesting new data. Methodologically the approach taken by the authors is generally sound. The conclusions that the authors make from their data also is logical and justified. I was particularly taken by the observation that the sodium window current plays a significant role in the enhancement of action potential firing. Judging the first two figures of the paper, there seems to be striking differences in the waveforms of action potentials recorded from Casr positive and negative neurons. Do the authors have an explanation for this, does for example Casr influence the developmental trajectory of neurons in culture, in terms of maturation of the expression of voltage gated ion channels? I think the manuscript would benefit from analysis that indicates at given stages of culture that current densities are equivalent etc, which would allow clearer comments to be made by the authors. My main problem with the work lies in the voltage-clamp analysis. The authors indicate in mature cultures that inward currents cannot be voltage-controlled. They therefore turn to analysis of neurons that have been cultured for only a few days. I am not sure that this is the right thing to do – as alluded to above. I would suggest that the authors consider employing recording approaches, such as nucleated patch, that would allow near perfect voltage control. This would allow analysis at the same developmental stage, and also if made from various ages in culture the charting of the developmental profile.

We have performed the suggested new experiments using nucleated patches from neurons after >14 days in culture (Figure 3). These experiments address the concerns about VGSC currents being examined in neurons of different ages to those examined in other parts of the manuscript.

[Editors’ note: what follows is the authors’ response to the second round of review.]

Essential Revisions:As indicated in the public comments, the authors need to systematically reword the presentation, starting with the title, to reflect the fact that all their experiments examined effects of reducing both Ca and Mg, and it is therefore incorrect to state the changes reflect "[Ca^2+^]-dependent" effects as is done throughout. There were also many places where the wording (and sometimes the order of presentation) seemed unnecessarily convoluted, which made it difficult to understand exactly what the authors meant by a particular sentence. The data in the paper are very interesting and the ability of the reader to digest the results and their meaning would be improved if the presentation were clearer in some places.Following points in particular require careful consideration:Title: "CaSR modulates sodium channel-mediated Ca^2+^-dependent excitability". The title is misleading in two ways. First, the main point of the paper is that the most important effects of reducing divalents are NOT mediated by CaSR receptors. Whatever effects on excitability are seen in the CaSR-/- neurons probably reflect homeostatic changes rather than direct function of CaSR receptors, so even the statement that CaSR might "modulate" effects is questionable. In any case, such modulation is a very much secondary point of the paper, not the main one. Second, as already pointed out, the authors looked at changes in response to changing both Ca and Mg and can't say any of the key phenomena examined in the paper are "Ca^2+^-dependent". Something like "Enhanced excitability of cultured cortical neurons in low-divalent solutions is mediated mainly by altered voltage-dependence of voltage-dependent sodium channels" would better capture the content and conclusions of the paper.

We have modified the title to better represent the papers major findings. We have used:

Enhanced excitability of cortical neurons in low-divalent solutions is primarily mediated by altered voltage-dependence of voltage-gated sodium channels

Abstract: Abstract was not understandable until after reading the paper. The authors summarize the key results as follows " Here we report that action potential (AP) firing rates increased in wild-type (WT), but not CaSR null mutant (Casr-/-) neocortical neurons, following the switch from physiological to reduced Ca ^2+^-containing Tyrode. However, after membrane potential correction, action potential firing increased similarly in both genotypes inconsistent with CaSR regulation of NALCN." It was completely unclear whether the authors were saying that reduction of calcium did or did not enhance excitability in the CaSR-/- cells, the central question of the paper. The first sentence says "no" and the second sentence says "yes", but only after a "membrane potential correction" that is completely undefined. It is also completely unclear until after reading the paper that "action potential firing" refers to spontaneous firing at the resting potential. As it turns out, a clearer description of the results would be something along the lines of: "CaSR-/- neurons had more negative resting potentials than control neurons and unlike control neurons did not fire spontaneously either in normal or reduced calcium solutions. However action potential firing in response to depolarizing current injections was enhanced in CaSR-/- neurons by reduction of divalent cations to a similar degree as in control neurons."

We agree that the abstract was unclear. We have adopted many of the reviewers’ suggestions and believe they have helped clarify the abstract.

p. 3 "Classical studies proposed that the mechanism underlying [Ca^2+^]o-dependent excitability centers on voltage-gated sodium channel (VGSC) sensitivity to extracellular Ca^2+^. External Ca^2+^ was proposed to interact with local negative charges on the extracellular face of the membrane or ion channels thereby increasing the potential field experienced by VGSCs and reducing the likelihood of VGSC activation at the resting membrane potential (Frankenhaeuser and Hodgkin, 1957; Hille, 1968)". Frankenhaeuser and Hodgkin only looked at shifts of Na channel gating in voltage clamp experiments and did not report firing of the cells at the resting potential, as the reader might conclude from this. In fact in their discussion they conclude because of the shift in inactivation, axons in low Ca would be inexcitable. A more accurate summary might be something like "Reduced external Ca^2+^ was proposed to shift the effective voltage-dependent gating of the sodium conductance in the hyperpolarizing direction by reducing the screening of local negative charges on the extracellular face of the membrane by external Ca^2+^ (Frankenhaeuser and Hodgkin, 1957; Hille, 1968)".

Our approach was taken to try and emphasize the action of calcium rather than the effect of removing calcium but we agree the reviewers’ version is clearer and have changed the text. We have attempted to clarify what we mean by excitability and have drawn on the work of Frankenhaeuser and Hodgkin. In their introduction, Frankenhaeuser and Hodgkin point out that they investigated the mechanism responsible for excitability including spontaneous firing. ”Physiologists have been interested in the action of calcium on excitable tissues since the days of Ringer (1883). Some of the main facts established (see Brink,1954) are that increasing the external calcium concentration raises the threshold, increases membrane resistance (Cole, 1949) and accelerates accommodation. Reducing the calcium concentration has the converse effects, and frequently leads to spontaneous oscillations or repetitive activity (e.g. Adrian & Gelfan, 1933; Arvanitaki, 1939).” While their approach focused on the voltage clamp method they did describe how low calcium increased excitability “Another advantage of using anodal polarization was that it stopped the fibre firing repetitively in low calcium solutions.” As far as we can determine they reported that reduced excitability was only seen after prolonged exposure to zero calcium conditions and this is generally presumed to reflect a different mechanism, “Unpublished experiments with a conventional type of internal electrode indicate that although there may be little change in resting potential, squid fibres become inexcitable in zero calcium within 5-20 min.”

p. 11: "[Ca^2+^]o reduction and CaSR hyperpolarized this region of overlap towards the RMP (Figure 3 F,G insets) increasing the likelihood that persistent VGSC currents were activated at resting membrane potential and therefore contributing to [Ca^2+^]o-dependent excitability."The rationale for saying that [Ca^2+^]o reduction (or really divalent reduction) hyperpolarized the region of overlap is clear. However, the rationale connecting the experimental results with saying that "CaSR hyperpolarized the region of overlap" is very vague.

We have changed the section to improve clarity. It now says “ Divalent reduction hyperpolarized this region of overlap towards the RMP (Figure 3 F,G insets) increasing the likelihood that persistent VGSC currents were activated at resting membrane potential and therefore contributing to divalent-dependent excitability. The depolarization of VGSC activation gating that resulted from CaSR deletion (Figure 3I), shifted the area of conductance curve overlap for T_0.2_ in a depolarizing direction (Figure 3G).”

p. 14 paragraph heading "VGSCs are the dominant contributor to [Ca^2+^]o-dependent excitability". This statement seems almost meaningless. There is no excitability of any kind without VGSCs. The meaning of "[Ca^2+^]o-dependent excitability" here (and it is used many other times) is not clear. Perhaps the authors mean "changes in excitability produced by changes in divalent concentrations". In any case, the paragraphs to follow are focused mainly on how changes in resting potential are mediated, not really on excitability. So the meaning is something like "Changes in resting potential resulting from lowered divalents are mediated mainly by VGSCs".

I agree about the heading and have changed accordingly. We agree that we have used "[Ca^2+^]o-dependent excitability" to mean "changes in excitability produced by changes in divalent concentrations". We will clarify in the text.

p. 16: "We have investigated the mechanisms underlying [Ca^2+^]o-dependent changes in intrinsic neuronal excitability and tested if CaSR is transducing decreases in [Ca^2+^]o into NALCN-mediated depolarizations to trigger action potentials (Lu et al., 2010; Philippart and Khaliq, 2018). We found no evidence that this specific mechanism was active in neocortical neurons (Figure 2)."

I agree about the heading and have changed accordingly. We agree that we have used "[Ca^2+^]o-dependent excitability" to mean "changes in excitability produced by changes in divalent concentrations". We will clarify in the text.

Nothing in the Philippart and Khaliq paper examined whether or not CaSRs were involved in control of NALCN by divalents. Their paper concerned a completely different regulation of NALCN current by GABAB and dopamine receptors.

We agree that Philipart and Khaliq did not study CaSR regulation of NALCN and have deleted the citation.

p. 16 "CaSR also indirectly affected action potential generation by modestly depolarizing the membrane potential (Figure 1)." The authors have no evidence that stimulating CaSRs depolarizes the resting potential. What they mean is that CaSR-/- neurons have hyperpolarized resting potentials compared to WT neurons. They have no evidence of whether this reflects a loss of a normal depolarizing effect of CaSR stimulation or a homeostatic effect of development of the neurons without CaSRs.

We agree and have modified the text accordingly.

p. 21 "Voltages have been corrected for liquid junction potentials." The value used for the corrections should be given, as the exact membrane potentials are quite important for the effects that are shown (especially the firing at the resting potential).The switch from the use of CreWT to ControlWT lacking the expression of Cre recombinase still remains to be justified. The authors reason that the switch has been made to ControlWT given that they have shown that calcium-dependence of excitability is not mediated by CaSR-modulation of NALCN but by VGSC, and also in order to avoid any confounds presented by the exogenous expression of Cre. Nevertheless, the logic is not clear. The conclusion of the apparent lack of contribution of CaSR-modulation of NALCN has been reached by the use of Cre expressing neurons. The authors should explicitly state whether or not there were any differences in the electrophysiological properties of conWT and creWT neurons and if so, explain what they are, perhaps in the methods.

We have addressed the rationale for use of WT and added it to the methods section as suggested. We did not identify any differences in the electrophysiological properties of conWT and creWT neurons. In Figure 1 A-D we showed that the basal firing rates of the con WT and creWT responded the same to reduced divalent concentrations. We opted to use the creWT and *Casr^-/-^* neurons to examine if Casr was responsible for the sensitivity to extracellular divalents. Deletion of Casr changed sensitivity to divalents because of its impact on RMP and VGSC activation (V_0.5_). This comparison avoided possible confounding due to cre. Since we had determined that CaSR did not regulate NALCN and that the changes in excitability produced by changes in divalent concentrations were the same in cre and conWT neurons we switched back to conventional neurons. The new question we addressed was, what is the relative contribution of NALCN and VGSC to the depolarizations that occurred after switching to T_0.2_ and caused increased action potential firing? Since this question did not include the CaSR we used the conventional WT to ensure the results were generalizable. Our findings that VGSC currents mediated the majority of the changes in basal current and membrane potential following decreases in divalent concentrations means that even if CaSR had regulated NALCN, as hypothesized by Lu et al., the relative size of the effect would have been quite modest.

We were very surprised that most of the divalent mediated depolarization was sensitive to TTX in these WT neurons which also indicates NALCN is not having a substantial role in the mediation of the divalent dependent depolarization. The *Casr^-/-^* neurons are very similar as shown in the supplementary figures.

There were many typos in the paper.Throughout the text, there seemed to be a global replacement of "T1.1" by a space, which made a lot of sentences hard to read. For example, p. 12 " Switching to plus TTX produced minimal change in the basal current on average (Figure 5C). However, in some neurons elicited an outward current (Figure 5A,C) whereas in others there was an inward current (Figure 5B,C)…" It is assumed that there are two missing "T1.1"'s.p.11 D,E). The reduction in [Ca^2+^]o (F (1, 18) = 56, P < 0.0001) left- shifted V0.5 (2-way RM ANOVA,

We have corrected the typos and apologize for our error in replacing T1.1 with space.